# How the dynamic interplay of cortico-basal ganglia-thalamic pathways shapes the time course of deliberation and commitment

Zhuojun Yu[1], Timothy Verstynen[1,2]☺*, Jonathan E. Rubin[2,3]☺*

1 Department of Psychology & Neuroscience Institute, Carnegie Mellon University, Pittsburgh, Pennsylvania, United States of America, 2 Center for the Neural Basis of Cognition, Carnegie Mellon University & University of Pittsburgh, Pittsburgh, Pennsylvania, United States of America, 3 Department of Mathematics, University of Pittsburgh, Pittsburgh, Pennsylvania, United States of America

☺ These authors contributed equally to this work.
* timothyv@andrew.cmu.edu (TV); jonrubin@pitt.edu (JER)

## Abstract

Although the cortico-basal ganglia-thalamic (CBGT) network is identified as a central circuit for decision-making, the dynamic interplay of multiple control pathways within this network in shaping decision trajectories remains poorly understood. Here we develop and apply a novel computational framework—CLAW (Circuit Logic Assessed via Walks)—for tracing the instantaneous flow of neural activity as it progresses through CBGT networks engaged in a virtual decision-making task. Our CLAW analysis reveals that the complex dynamics of network activity is functionally dissectible into two critical phases: deliberation and commitment. These two phases are governed by distinct contributions of underlying CBGT pathways, with indirect and pallidostriatal pathways influencing deliberation, while the direct pathway drives action commitment. We translate CBGT dynamics into the evolution of decision-related policies, based on three previously identified control ensembles (responsiveness, pliancy, and choice) that encapsulate the relationship between CBGT activity and the evidence accumulation process. Our results demonstrate two contrasting strategies for decision-making. Fast decisions, with direct pathway dominance, feature an early response in both boundary height and drift rate, leading to a rapid collapse of decision boundaries and a clear directional bias. In contrast, slow decisions, driven by indirect and pallidostriatal pathway dominance, involve delayed changes in both decision policy parameters, allowing for an extended period of deliberation before commitment to an action. These analyses provide important insights into how the CBGT circuitry can be tuned to adopt various decision strategies and how the decision-making process unfolds within each regime.

**Data availability statement:** The network codebase utilized in this study is publicly available at https://github.com/CoAxLab/CBGTPy. Detailed installation instructions and a comprehensive list of implemented functions can be found in the README.txt file within the repository. All datasets generated and analyzed during the course of this research, along with a demonstration demo, is publicly available at https://github.com/zhuojunyu-appliedmath/CLAW.

**Funding:** TV and JER received funding from the National Institutes of Health award R01DA059993 (https://www.nih.gov) as part of the CRCNS program (https://www.nsf.gov/funding/opportunities/crcns-collaborative-research-computational-neuroscience). JER was also partly supported by the NIH award R01NS125814, also part of the CRCNS program. The funders had no role in study design, data collection and analysis, decision to publish, or preparation of the manuscript.

**Competing interests:** The authors have declared that no competing interests exist.

**Abbreviations:** CBGT, cortico-basal ganglia-thalamic; CCA, canonical correlation analysis; CLAW, Circuit Logic Assessed via Walks; Cx, cortical neurons; CxI, cortical interneurons; DDM, drift-diffusion model; dSPNs, direct spiny projection neurons; FSI, fast-spiking interneurons; GPe, external globus pallidus; HSSM, Hierarchical Sequential Sampling Modeling; iSPNs, indirect spiny projection neurons; KL, Kullback–Leibler; PS, pre-stimulated state; STN, subthalamic nucleus; Th, thalamus.

## Author summary

We investigate how the cortico-basal ganglia-thalamic (CBGT) network coordinates decision-making through its interconnected pathways. Using a novel Circuit Logic Assessed via Walks (CLAW) framework, we trace instantaneous neural activity through virtual CBGT networks as they engage in forced choice decisions. This analysis uncovers two key phases of a decision: deliberation, shaped by the indirect and pallidostriatal pathways, and commitment, driven by the direct pathway. We also demonstrate that CBGT activity supports two distinct decision styles: fast decisions involve an early decision boundary collapse and strong directional preference, while slow decisions feature minimal changes during an extended deliberation phase. These findings reveal the dynamic mechanisms within the CBGT network that underlie the different decision processes and how these can be tuned to adapt decision-making across varying demands and contexts.

## Introduction

Theoretical [1–5] and empirical [6–9] studies have established that the basal ganglia, working together with the thalamus and cortex, can play a critical role in the evidence accumulation process during decision-making. The distributed circuits of the cortico-basal ganglia-thalamic (CBGT) network simultaneously integrate external (e.g., sensory signals) and internal (e.g., learned contingencies) factors, until sufficient evidence is accumulated to allow one action to proceed [10,11]. The topological structure of the interconnected CBGT nuclei [12–14], including their spatiotemporal organization into separable action representations [15], is ideally suited for managing this evidence accumulation process [3,16,17] and adapting its implementation based on environmental feedback [12,18–21].

Prior computational work [5,22,23] has shown how the CBGT network can regulate decision policies (e.g., manage the speed-accuracy [24,25] and exploration-exploitation [26,27] tradeoffs). Specifically, interactions between distinct subnetworks within the CBGT pathways can be mapped to certain behavioral outcomes described by parameters in a drift-diffusion model (DDM; [10,16,28]), with the low-dimensional relationship between specific configurations of CBGT networks and decision policies characterized in terms of three *control ensembles* [22]. It has been further shown that dopaminergic plasticity at the corticostriatal synapses can alter the activity of CBGT control ensembles, so as to modulate the resulting decision policies in response to feedback following post-action outcomes [23]. The parameters of the classical DDM are typically interpreted as time-invariant quantities, estimated across *multiple* trials to capture a fixed decision policy during a decision. Thus, the previous control ensemble studies suggest that the CBGT network can establish a *static* instantiation of such a policy. In typical evidence accumulation models this would be represented by fixed values of parameters such as drift rate (the rate of evidence accumulation

toward a decision) and boundary height (the amount of evidence needed to make a decision), with gradual adaptation of these factors over the course of sequences of decisions if feedback is provided. However, it remains to be investigated how evidence accumulates through different regions of the CBGT network during the course of *individual* decisions.

A number of studies have recently challenged the assumption of static decision policies from both experimental [29–32] and theoretical [33–35] perspectives, showing that the classical DDM fails to capture important trends in behavioral data under some specific tasks. The intuition behind these studies is that the decision-makers adjust their decision strategies based on real-time demands. In line with this idea, there is a growing body of literature [36–40] that explores the idea of *dynamic* evidence accumulation, in which certain parameters, particularly drift rate and boundary height, are allowed to vary within a trial to capture more realistic, temporally evolving decision policies. This raises our second question: how do the complex dynamics of CBGT networks give rise to rapid, within-trial variations in the decision policy?

To address the first of these open questions, we propose in this work a novel computational framework that we call CLAW, short for Circuit Logic Assessed via Walks, to depict the flow of noisy neural activity through a model CBGT network on a moment-by-moment basis (Fig 1A–D). Our CLAW analysis is related to, but extends, the classical framework for conceptually organizing CBGT activity, which assumes that the internal circuitry of the basal ganglia implements action selection through a division of labor between the structurally and functionally dissociable *direct* pathway (Fig 1A, green) and *indirect* pathway (blue) [13,41,42], with the former facilitating action selection and the latter suppressing action selection [41]. Discoveries over the past decade have revealed that, rather than functioning as independent and antagonistic mechanisms for facilitating or suppressing action selection, the direct and indirect pathways may engage in a dynamic competition for control over basal ganglia output [3,17,43,44]. Moreover, this model of CBGT pathways has been further complicated by a recent reappreciation for the cellular complexity of the external segment of the globus pallidus (GPe; see [45]), now seen as being composed of two general classes of neurons: prototypic (GPeP) and arkypallidal (GPeA) cells [46–49]. The arkypallidal neurons are currently thought to play a critical role in regulating striatal signaling, particularly during tasks requiring reactive inhibitory control [50–53]. In light of these findings, here we propose a computational model that incorporates an additional third pathway, the *pallidostriatal* pathway (Fig 1A, gold), that conveys ascending feedback inhibition to the striatum and is modulated by both striatal projection pathways. Then applying the CLAW framework to simulated CBGT dynamics in our model specifically allows us to generate novel insights and predictions about the complex interactions among all CBGT nuclei during decision processes, helping us to extract the details of how specific pathways interact to control decision-making dynamics. Here we use this approach to explore the way the direct, indirect, and pallidostriatal pathways regulate the bidirectional control of information through CBGT circuits and hence the dynamics of evidence accumulation as a decision is being made.

To address the second question, about the dynamic variations of decision policies within the fast timescale of individual decisions, we build on the prior observation of separable control ensembles within the CBGT pathways [22,23]. These groupings of basal ganglia components, dubbed the *responsiveness*, *pliancy*, and *choice* ensembles, each represent a relation between CBGT firing patterns and DDM parameters that encode a decision policy. Responsiveness modulates how quickly evidence evaluation begins and the standard of evidence needed for decisions, in a positively correlated way, and is largely associated with the overall activity in the corticothalamic loop and direct pathway. Pliancy modulates evidence onset and the amount of evidence required to make a decision in opposite ways, and is strongly dependent on the overall activity of the indirect pathway. Choice modulates the direction and speed of evidence accumulation towards one action. With this ensemble the overall activity across channels has very little impact on the resulting decision policy, but instead the between-channel differences in firing activity show a robust association. This control ensemble framework provides mechanistic predictions about the roles of CBGT pathways in tuning decision-related policies [22], and how dopamine-mediated synaptic plasticity gradually shifts decision properties across multiple decisions [23]. Yet until now, the detailed dynamics of how these control ensembles shape information flow during the decision process has remained unknown.

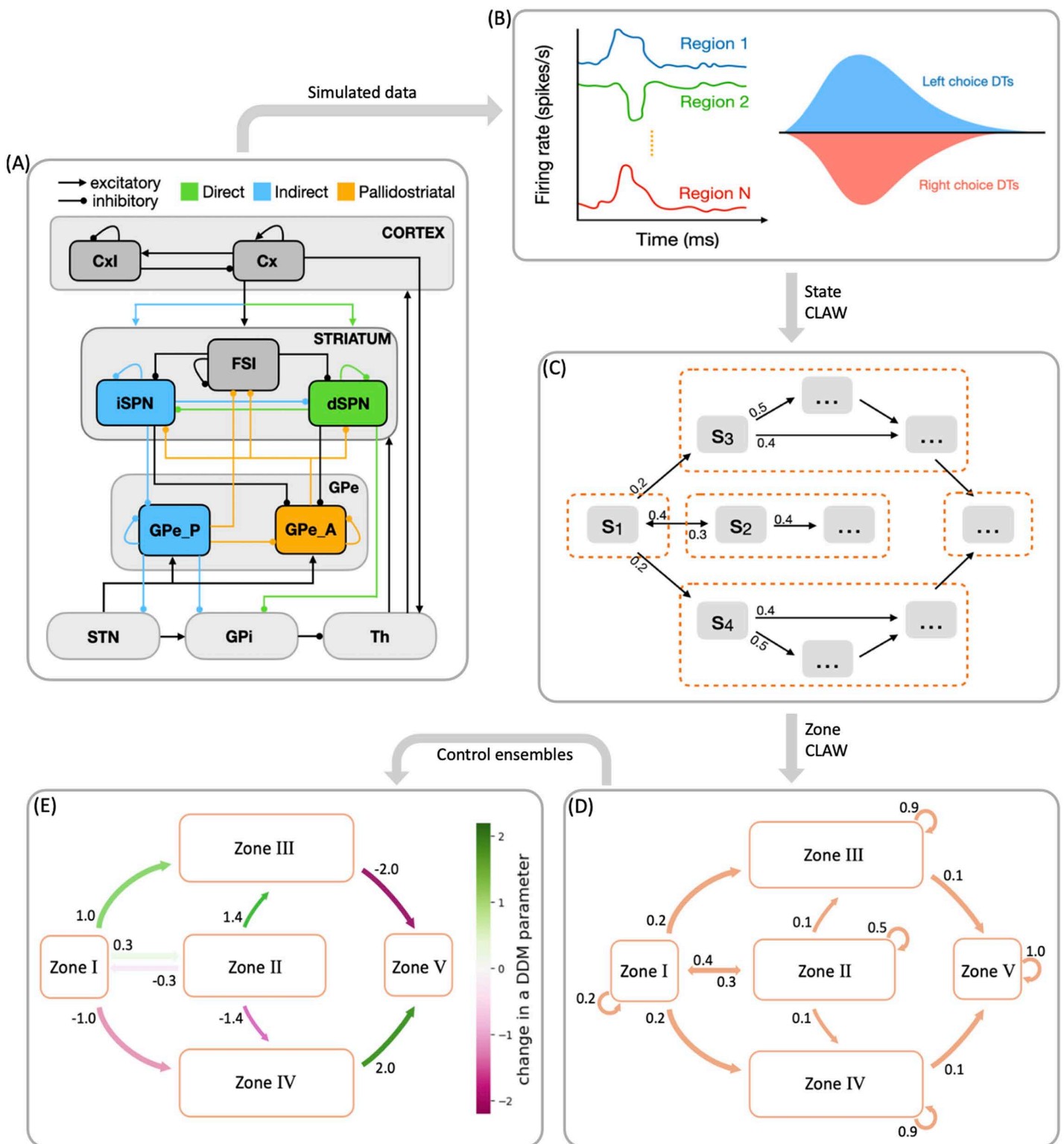

**Fig 1. Schematic showing the steps in analyzing decision-making dynamics within the cortical-basal ganglia-thalamic (CBGT) network. (A)** CBGT network, with connections color-coded as follows: classical direct pathway in green, classical indirect pathway in blue, and pallidostriatal pathway in gold. Arrows ending in dots indicate postsynaptic sites of inhibitory connections, while arrows ending in triangles indicate excitatory connections. Cx,

cortical neurons; CxI, inhibitory cortical interneurons; FSI, fast spiking interneurons; dSPN, direct spiny projection neurons; iSPN, indirect spiny projection neurons; GPe, external globus pallidus; GPe_P, prototypical neurons; GPe_A, arkypallidal neurons; GPi, internal globus pallidus; STN, subthalamic nucleus; Th, thalamus. **(B)** Simulated behavioral data, including firing rates, choices, and decision times, generated by a spiking model for the CBGT network. **(C)** A chain of state transitions derived from the processed data. **(D)** States grouped into zones, with associated transition probabilities. **(E)** Levels of activity within zones mapped to DDM parameters through control ensemble analysis, showing how dynamic decision policies emerge from flow of network activity between zones.

In this work we set out to replicate the existence of the three independent control ensembles, using the more biologically realistic model of CBGT circuits that includes the pallidostriatal pathway for additional control. We then try to translate firing rate changes associated with transitions between different network states to changes in the DDM parameter space (as sketched in Fig 1D and 1E), allowing us to predict dynamic fluctuations and adjustments in decision policies over the fast timescale on which individual decisions are made. Our results reveal how instantaneous information flow through different CBGT pathways produces a dynamic decision policy and demonstrate two contrasting decision-making modes: fast decisions feature an early collapse in the evidence accumulation threshold and a clear directional bias, while slow decisions with more extended deliberation are driven by a delayed response in the decision boundary height and drift rate. By comparing the dynamics of control ensemble activation across fast and slow trials, we gain important insights about the contributions of specific aspects of CBGT activity to flexible decision-making, which may be harnessed to adapt decision policies during the evidence accumulation process.

## Results

### CBGT activity and CLAW

Our main goal here was to extract the temporal dynamics of information flow through CBGT circuits during the evidence accumulation process within individual decisions. To this end, we simulated a spiking model of the CBGT network [54] in the context of a simple two-choice task. Implementing a genetic search algorithm [23] we generated 300 distinct networks, each with a different configuration of synaptic weights, that produced firing rates of all CBGT populations within experimentally observed ranges (see also [22]). The networks were largely unbiased with respect to left and right choices, as the same connectivity parameters were shared by both channels. For each sampled network, we simulated 50 trials (i.e., choices) and gathered the time-dependent firing rates from a collection of $N = 10$ distinct cell populations within the CBGT network, across action channels representing left and right choices (Fig 1B). As a reference, Fig 2 shows an example of the time course of firing rates for all the nuclei in two consecutive trials from an example network. The firing rates were tracked up to the decision time (DT; Fig 2, pink region), defined as the time at which the instantaneous firing rate of the thalamic population for either of the channels first reaches a pre-specified decision threshold (set to 30 Hz). The network firing rates were computed in bins of width $\Delta t = 10$ ms and binarized based on whether activity in each bin was above or below a predefined threshold (Fig 2, green horizontal lines). We then defined a *state* $s_k \in \mathbb{R}^N$ ($k = 1, 2, 3, \cdots, 2^N$) to be a unique pattern of activity across the $N$ nuclei of interest in our network. Specifically, each $s_k = [\sigma_{k1}, \sigma_{k2}, \cdots, \sigma_{kN}]$, where $\sigma_{kj} \in \{0, 1\}$. Thus, for each trial, we can convert the sequence of binarized firing rates into a sequence of states, each representing the pattern of CBGT activity in a time window. By aggregating all of the state sequences, we determine the transition probabilities between states and then construct a chain that describes the flow of neural states that occurs across the simulated decision processes (Fig 1C). See CBGT network in the Methods section for details of our modeling framework.

High-probability states within the CBGT network during the evidence accumulation process were analyzed using a novel procedure that we call *Circuit Logic Assessed via Walks* (CLAW). The CLAW diagram is depicted in Fig 3A. The name derives from the fact that this analysis can uncover how activity flows through the CBGT circuit by looking at how different cell populations are engaged together over the course of a decision. Transitions between states resemble walks,

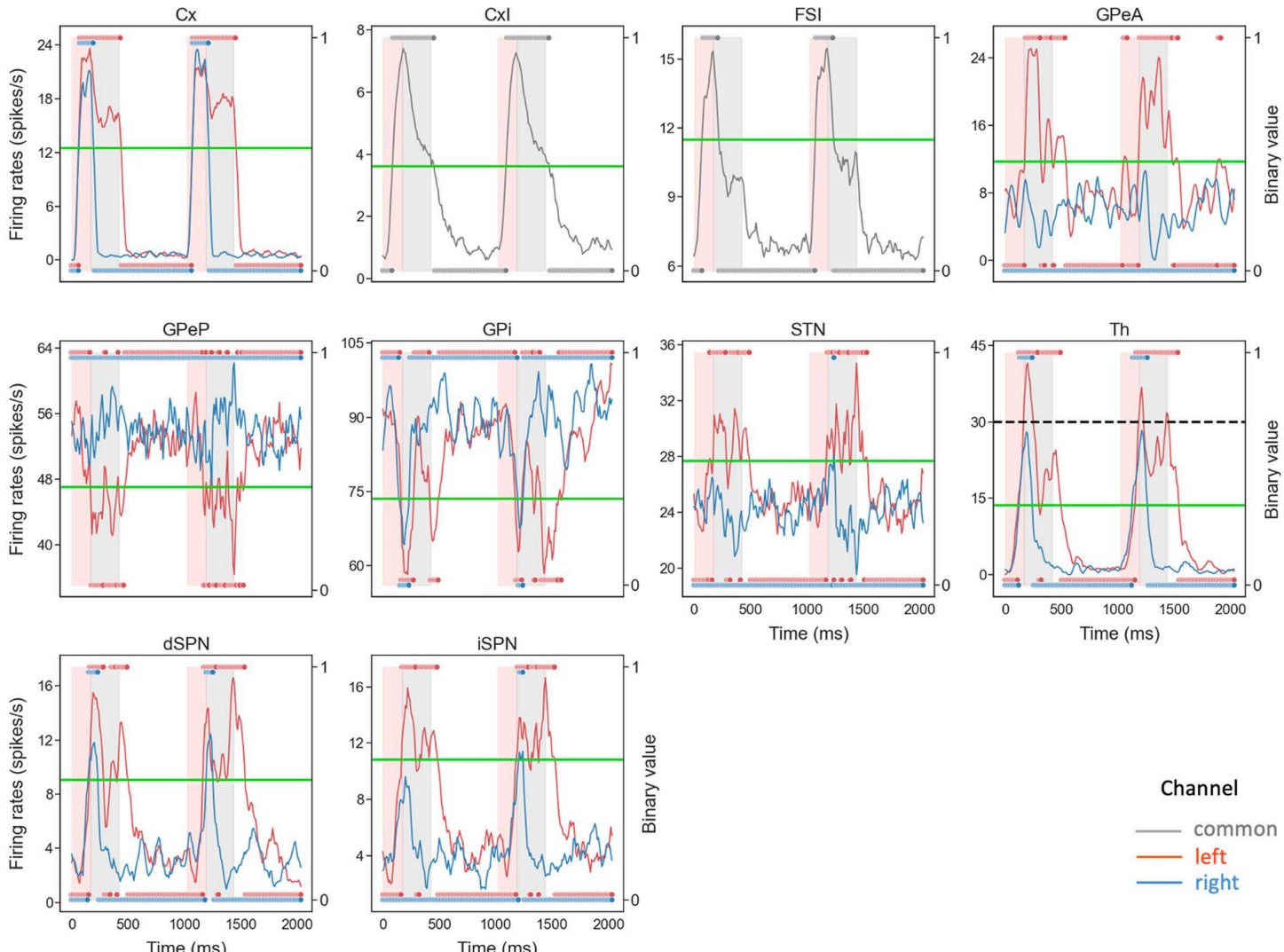

**Fig 2. Example of firing rate time course and binarization in a two-choice task.** Each red (blue) trace corresponds to activity in the left (right) action channel in a CBGT region, and the grey traces correspond to the populations (i.e., CxI and FSI) common to both action channels. Pink regions represent the decision-making phase, occurring before the thalamus (Th) of one of the action channels reaches the decision threshold of 30 Hz (dashed black line in Th panel). Grey regions represent the consolidation phase, where partial cortical input to the selected channel is sustained [55,56]. The unshaded regions represent the inter-trial interval. In each panel the right *y*-axis corresponds to binarized firing rates (dots at 0 or 1), where the horizontal green line indicates the binarization threshold (see Fig 10 for details on how thresholds were determined).

or chains that can revisit the same state, rather than paths, where each state is visited only once [57]. These transition probabilities from the current state to a subsequent state of the CLAW are shown on the directed edges in Fig 3A. We also show how each state relates to the speed characteristics of a decision by color-coding each state with the mean DT of the trials that visited that state. For purposes we elaborate on later, we further divided the decision time distribution of the full set of simulated networks into tertiles of equal mass, respectively defining fast (short DT), intermediate (medium DT), and slow (long DT) networks (Fig 3B). Finally, the table in Fig 3C shows the details of the activity associated with each CLAW state, including the binarized firing rates of the populations of interest, the activation probability of subthalamic nucleus

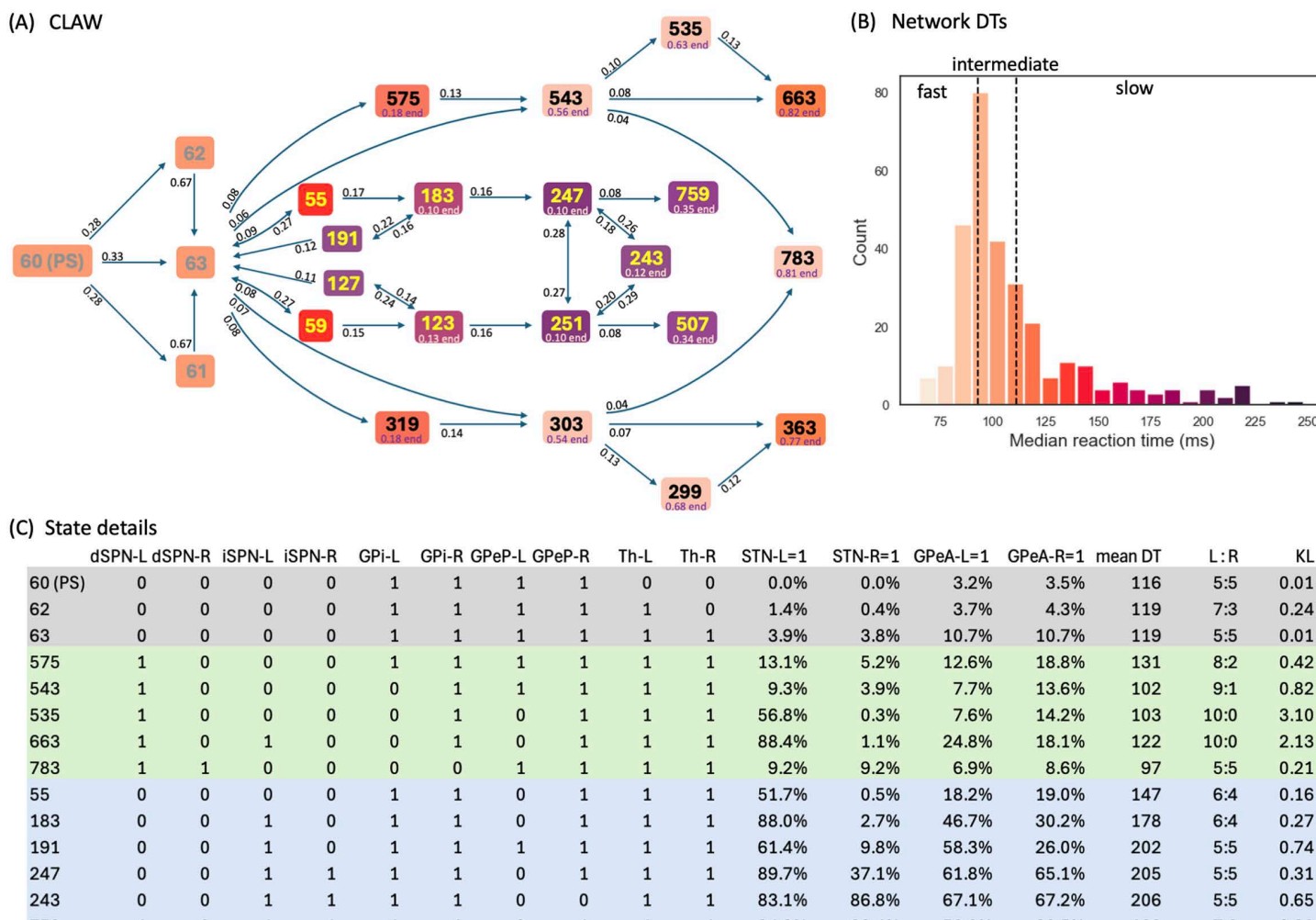

**(A)  CLAW**

**(B)  Network DTs**

**(C)  State details**

| | dSPN-L | dSPN-R | iSPN-L | iSPN-R | GPi-L | GPi-R | GPeP-L | GPeP-R | Th-L | Th-R | STN-L=1 | STN-R=1 | GPeA-L=1 | GPeA-R=1 | mean DT | L:R | KL |
|---|---|---|---|---|---|---|---|---|---|---|---|---|---|---|---|---|---|
| 60 (PS) | 0 | 0 | 0 | 0 | 1 | 1 | 1 | 1 | 0 | 0 | 0.0% | 0.0% | 3.2% | 3.5% | 116 | 5:5 | 0.01 |
| 62 | 0 | 0 | 0 | 0 | 1 | 1 | 1 | 1 | 1 | 0 | 1.4% | 0.4% | 3.7% | 4.3% | 119 | 7:3 | 0.24 |
| 63 | 0 | 0 | 0 | 0 | 1 | 1 | 1 | 1 | 1 | 1 | 3.9% | 3.8% | 10.7% | 10.7% | 119 | 5:5 | 0.01 |
| 575 | 1 | 0 | 0 | 0 | 1 | 1 | 1 | 1 | 1 | 1 | 13.1% | 5.2% | 12.6% | 18.8% | 131 | 8:2 | 0.42 |
| 543 | 1 | 0 | 0 | 0 | 0 | 1 | 1 | 1 | 1 | 1 | 9.3% | 3.9% | 7.7% | 13.6% | 102 | 9:1 | 0.82 |
| 535 | 1 | 0 | 0 | 0 | 0 | 1 | 0 | 1 | 1 | 1 | 56.8% | 0.3% | 7.6% | 14.2% | 103 | 10:0 | 3.10 |
| 663 | 1 | 0 | 1 | 0 | 0 | 1 | 0 | 1 | 1 | 1 | 88.4% | 1.1% | 24.8% | 18.1% | 122 | 10:0 | 2.13 |
| 783 | 1 | 1 | 0 | 0 | 0 | 0 | 1 | 1 | 1 | 1 | 9.2% | 9.2% | 6.9% | 8.6% | 97 | 5:5 | 0.21 |
| 55 | 0 | 0 | 0 | 0 | 1 | 1 | 0 | 1 | 1 | 1 | 51.7% | 0.5% | 18.2% | 19.0% | 147 | 6:4 | 0.16 |
| 183 | 0 | 0 | 1 | 0 | 1 | 1 | 0 | 1 | 1 | 1 | 88.0% | 2.7% | 46.7% | 30.2% | 178 | 6:4 | 0.27 |
| 191 | 0 | 0 | 1 | 0 | 1 | 1 | 1 | 1 | 1 | 1 | 61.4% | 9.8% | 58.3% | 26.0% | 202 | 5:5 | 0.74 |
| 247 | 0 | 0 | 1 | 1 | 1 | 1 | 0 | 1 | 1 | 1 | 89.7% | 37.1% | 61.8% | 65.1% | 205 | 5:5 | 0.31 |
| 243 | 0 | 0 | 1 | 1 | 1 | 1 | 0 | 0 | 1 | 1 | 83.1% | 86.8% | 67.1% | 67.2% | 206 | 5:5 | 0.65 |
| 759 | 1 | 0 | 1 | 1 | 1 | 1 | 0 | 1 | 1 | 1 | 94.9% | 32.4% | 53.8% | 68.5% | 192 | 7:3 | 0.93 |

**Fig 3. CLAW (Circuit Logic Assessed via Walks) diagram for CBGT network dynamics. (A)** CLAW diagram. Bold numbers in boxes indicate the network states, and the transition probability from a current state to a subsequent state is indicated by the number near the arrow pointing from the current state. The numbers below certain states (e.g., "0.82 end" below state 663) represent the probability that if these states are reached, then they are the final states prior to decisions. **(B)** Overall decision time distribution across 300 networks, categorized into three equal-count tertiles defining fast (left), intermediate (middle), and slow (right) networks, demarcated by vertical dashed black lines. The coloring of each CLAW state in panel A corresponds to the mean DT for all trials that visit this state, following the same color-coding scheme as in panel **B**. **(C)** Details of the states. A complete explanation of the full set of state properties, including those related to the right choice, is presented in the Supporting Information S1 Table. From left to right, after state labels: binarized firing rates of dSPN, iSPN, GPi, GPeP, and Th for left (-L) and right (-R) channels; probability of activation (binarized firing rate = 1) for STN and GPeA for left and right channels; mean DT over the trials that visit each state; the ratio of trials that chose left/right for those that visit each state; Kullback–Leibler (KL) divergence between left and right trials' DT distributions. The grey rows correspond to the initial states that occur early in each trial and never lead directly to a decision, and the green and blue rows correspond to outer CLAW and inner CLAW states, respectively. The states in the lower half of the CLAW are not shown; these are symmetric – up to the swap of certain L and R channel binary values – with the states that lie in corresponding positions in the upper half.

(STN) and arkypallidal populations, the mean DT, the ratio of trials in which the left/right option was chosen, and the Kullback–Leibler (KL) divergence between the left and right trials' DT distributions.

Note that for our CLAW analysis we used an arbitrary numbering system that indicates the unique state out of all $2^{10}$ possible states of the network, focusing only on those that occurred with relatively high frequencies. Starting from the

decision onset state of the CLAW, i.e., the pre-stimulated state (PS) 60, either the left thalamus, the right thalamus, or both ramp up and cross the binarization threshold, progressing through states 61, 62, and 63, while the binarized activation states of the other nuclei remain unchanged. Note that at this point, thalamic firing typically remained below the decision threshold of 30 Hz, as this threshold is higher than the thalamic binarization threshold (see the Th panel in Fig 2), which was based on the probability distribution of firing rates for this nucleus. Consequently, evidence accumulation continues through changes in firing of the other CBGT components until an action is selected. Following the flow of transition probabilities in the CLAW diagram revealed two distinct paths of decision-making: the inner CLAW (states colored blue in Fig 3C), with low probabilities of transitioning directly into a decision, and the outer CLAW (states colored green), with high probabilities of transitioning directly into a decision.

We found two distinct loops along the inner CLAW, associated with deliberation: an initial deliberation phase represented by the loop $63 \rightarrow 55 \rightarrow 183 \rightarrow 191 \rightarrow 63$ (and symmetrically $63 \rightarrow 59 \rightarrow 123 \rightarrow 127 \rightarrow 63$), and a second deliberation phase represented by the loop $247 \leftrightarrow 243 \leftrightarrow 251 \leftrightarrow 247$. The initial deliberation phase provides the network with the flexibility to reconsider its current trajectory of evidence, allowing a trial to return to state 63 for a reassessment of the evidence and then reset the direction of information flow. The inherent uncertainty in this phase, as reflected by the balanced ratio of choosing left versus right at these states, indicates that the options are still being weighed at this stage. During the second deliberation phase, activity had a high probability of switching between three different states via changes in the activation of key nodes like the prototypical pallidal neurons, thereby potentially influencing the subsequent decision outcome (discussed below). This switching reflects a second-tier deliberative process in which alternative actions were actively considered before committing to a final choice.

In contrast to the flexible, evaluative paths along the inner CLAW, the outer CLAW paths were characterized by predominantly one-way transitions, leading unambiguously toward the final choice. In further support of the idea of distinct deliberative and commitment phases, we noted that the states within the inner CLAW displayed no clear preference between left and right choices, which suggests that the network was still exploring different possibilities without any strong bias. Conversely, trials that progressed along the outer CLAW paths exhibited increasing certainty and stronger commitment to the choice. For instance, the left-to-right (L:R) ratio increased along the one-way arrows from state 63 to states 575, 543, 535, and 663, ultimately reaching an absolute 10:0 at states 535 and 663, which signifies a firm commitment to the chosen alternative once the trial reached these states. Therefore, based on comparisons of the transition routes and outcome ratios between the inner and outer paths, the CLAW demonstrates that the complex dynamics of decision-making is separable, reflecting distinct states associated with deliberation and commitment. Different CBGT subnetworks have distinct activity patterns tied to these state transitions and hence distinct circuit logic underlying the decision process.

The above analysis raises questions about how specific pathways in the CBGT network control the decision dynamics. Our model incorporated three distinct pathways (subnetworks): the direct pathway (colored green in Fig 1A), the indirect pathway (blue), and the pallidostriatal pathway (gold). We observed that when the direct pathway, via D1-expressing direct spiny projection neurons (dSPNs) in the striatum, became dominant in an action channel (state 575 for left decisions), the process bypassed the two deliberative phases and rapidly committed to the corresponding action, leading to a short decision time characteristic of fast or intermediate networks. In the case of left decisions, the activation of the left dSPN inhibited the firing of the ipsilateral internal globus pallidus (GPi; 575 to 543), thus increasing left thalamic firing rate to the decision threshold (at states 543, 535, or 663). Notably, this route features either no, or relatively late, engagement of the indirect (including suppression of the GPeP) and pallidostriatal (including activation of the GPeA) pathways, indicating that the direct pathway was the primary driver of action commitment. However, if the indirect pathway, via D2-expressing indirect spiny projection neurons (iSPNs), turned on more quickly than the direct pathway (state 183), the trial spent more time in deliberation and had a longer decision time, as indicated by the darker coloring of the inner CLAW, corresponding to slow decision trials (compare colors in Fig 3A and 3B). Here the STN also became active (see "STN-L = 1" at state 183 in Fig 3C) due to the inhibitory influence of D2 striatal inputs onto the GPeP, which effectively released STN from inhibition. At the

same time, enhanced excitation from the STN and reduced inhibition from the GPeP gradually enhanced the firing of arky-pallidal neurons, especially when the trial was involved in the second deliberation loop (see "GPeA-L = 1" and "GPeA-R = 1" at states 243, 247, and 251). Thus, in these cases, the dominant indirect pathway actively induced pallidostriatal pathway activation, further contributing to a slowdown in decision-making. Note that in this scenario, the direct pathway was always suppressed by the dominant activity of the indirect SPNs and arkypallidal neurons, so that the trial was prevented from quickly executing a decision. Comparing the activation states of the CBGT nuclei and the DTs between the inner CLAW and the outer CLAW shows that the relative dominance of the CBGT pathways at any given time determines how quickly the system converges to a decision: the direct pathway accelerates decision-making while the indirect and pallidostriatal pathways decelerate it. Overall, although the CLAW approach extends beyond the classical CBGT pathways, we see that their distinct roles became evident within the framework of CLAW analysis.

We next explored the mechanistic details underlying the two deliberation phases. Early in the deliberation process, the decision trajectory could turn around from states 183 to 191, in which left prototypical GPe neurons returned to supra-threshold activity. This possibility resulted from the feedback loop between prototypical neurons and the STN: the suppression of activity in GPeP via its indirect striatal input disinhibited the STN, and then via the excitatory connections from STN to GPeP, the prototypical neurons in turn received excitatory input, which allowed them to overcome the inhibition from iSPNs and resume their supra-threshold firing. This state transition highlights the possible role of the reciprocal STN-GPe prototypical neuron loop in promoting exploration by lengthening the deliberation time for evidence accumulation. From state 191, the trajectory could bounce back to state 183 or evolve to state 63. The latter occured when the enhanced activity of arkypallidal neurons, which arose in state 191 due to the drive signals from STN, reduced the activity of indirect SPNs via the ascending pallidostriatal connections. This input halted the surge of decision-related striatal activity, including the potential activation of dSPNs, suppressed the planned action, and effectively reset the evidence evaluation process. From this process, we see the central role of the GPe, involving both prototypic and arkypallidal cell types, in regulating bidirectional information flow in order to implement the blocking of an incipient response.

On the other hand, if the trajectory did not turn around at state 183 but instead proceeded to state 247, a second deliberation phase could occur. Here, the battle between cortical drive and arkypallidal neurons' inhibition of direct striatal activity determined whether the trajectory was stuck in deliberation or committed to a decision. Importantly, no trial in our simulations failed to select an action, meaning that while the trial could spend time within the second deliberation phase, it did not remain perpetually in a state of deliberation. If the cortical drive became more intense, the ramping activity of the direct pathway (state 759 for the left channel and 507 for the right channel) led to greater commitment to the corresponding decision. In contrast, if dSPNs were inhibited by the stronger signals from arkypallidal neurons, then the indirect pathway could become dominant (states 247, 243, and 251) and temporarily suppress the selection of an action, with a descending influence on information flow to the GPeP, STN, GPi, and finally to the thalamus. Specifically, GPi output increased, providing an enhanced suppression of the thalamic responses and leading to a temporarily indecisive state. Note that the STN-GPeP subcircuit was also engaged during this deliberation phase, as demonstrated by the three-state reciprocal transitions between states 247, 243, and 251, where the activity of prototypical neurons in the two channels switched on and off. The competition between the left and right prototypical neurons determined the direction of the subsequent possible commitment, and if they were both sufficiently suppressed at state 243, then the deliberation time would be lengthened. From this phase, it is clear that the interactive effect of the indirect and pallidostriatal pathways played a critical role in prolonging the evidence accumulation process through the combination of a pause process at the striatal level along with downstream effects that allowed for switching between the two possible options.

To reduce the impact of firing rate variability on individual states and to categorize key classes of decision trajectories, we finally partitioned all CLAW states into six zones, as shown in Fig 4. The partition was based on the probabilities and activity changes associated with transitions across different states. Specifically, zone I contained the pre-stimulated state 60 as well as states 61, 62, and 63, where only thalamic populations showed a response in their binarized firing rates to

## (A) Zone CLAW

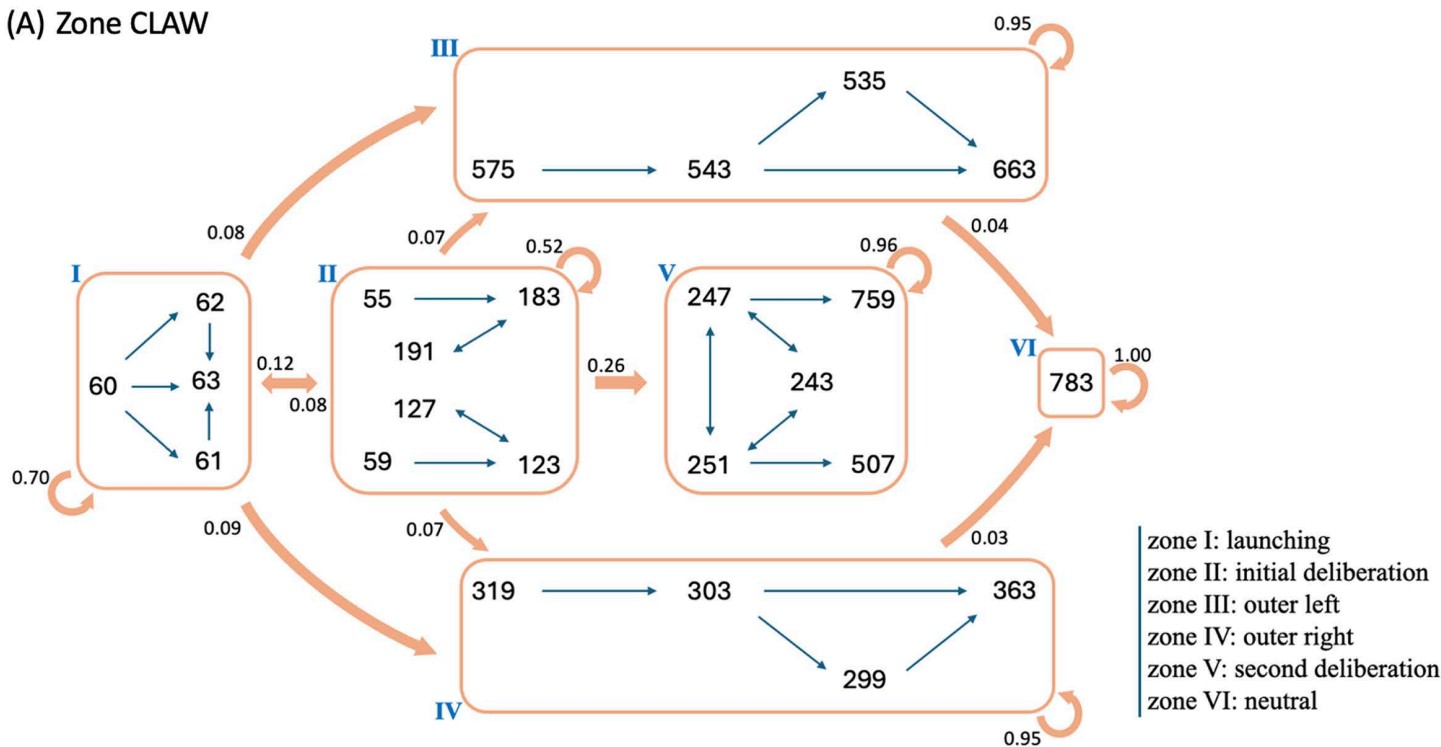

## (B) Zone details

|       | dSPN-L | dSPN-R | iSPN-L | iSPN-R | GPi-L | GPi-R | GPeP-L | GPeP-R | Th-L | Th-R | GPeA-L | GPeA-R | STN-L | STN-R |
|-------|--------|--------|--------|--------|-------|-------|--------|--------|------|------|--------|--------|-------|-------|
| I     | 0      | 0      | 0      | 0      | 1     | 1     | 1      | 1      | 0.23 | 0.23 | 0.05   | 0.05   | 0     | 0     |
| II    | 0      | 0      | 0.25   | 0.26   | 1     | 1     | 0.57   | 0.57   | 1    | 1    | 0.30   | 0.31   | 0.35  | 0.35  |
| III   | 1      | 0      | 0.14   | 0      | 0.32  | 1     | 0.71   | 1      | 1    | 1    | 0.12   | 0.16   | 0.29  | 0.03  |
| IV    | 0      | 1      | 0      | 0.13   | 1     | 0.32  | 1      | 0.71   | 1    | 1    | 0.15   | 0.11   | 0.03  | 0.29  |
| V     | 0.16   | 0.17   | 1      | 1      | 1     | 1     | 0.43   | 0.43   | 1    | 1    | 0.64   | 0.63   | 0.66  | 0.68  |
| VI    | 1      | 1      | 0      | 0      | 0     | 0     | 1      | 1      | 1    | 1    | 0.07   | 0.09   | 0.09  | 0.09  |

**Fig 4. CLAW partitioned into zones. (A)** Zone CLAW diagram. Zone I contains the pre-stimulated state 60 as well as states 61, 62, and 63, where only the binarized firing rates of thalamic populations may cross threshold. Zones II and V correspond to the initial and second deliberation phases within the inner CLAW, respectively. The left and right arms of the outer CLAW are represented by zones III and IV, respectively, each of which has a low probability of transitioning into zone VI, which consists of a single neutral state 783. The transition probability from one zone to another is indicated by the number near the arrow pointing from the source zone. The loop arrows represent the probability of staying in a zone or reaching the decision threshold from that zone. **(B)** Details of the zones. From left to right, after zone labels: probability of activation for dSPN, iSPN, GPi, GPeP, Th, GPeA, and STN for left (-L) and right (-R) channels, when networks are in each zone. The grey rows correspond to the launching zone, and the green and blue rows correspond to the zones containing the outer CLAW and inner CLAW states, respectively.

the stimulus at cortex. This zone serves as a pre-decision or launching zone, from which all trials originate, marking the initiation of the decision-making process. Zone II contained the group of states that could have a chance of returning to the pre-decision zone through the initial deliberation process. This inner CLAW zone played a critical role in early evaluation, where the system could explore alternatives rather than immediately converging on a specific action. In contrast, zones III and IV respectively corresponded to the left and right arms of the outer CLAW, each representing a strong commitment to a given action. Once the trial progressed into these zones, there was no return to deliberation and the

system was effectively locked into a specific decision. The only exception was that both outer CLAW zones could, with a low probability, transition to the more neutral state 783, which we defined as zone VI. In this state there was an equal likelihood of choosing left or right actions (L:R = 5:5). Such transitions occurred when enough competing evidence arrived to engage the contralateral direct SPNs before a decision was made. Lastly, zone V included the states that comprised the second deliberation process. We identified zones III, IV, V, and VI as *absorption zones*, in which the probability of staying until the decision threshold was reached was approximately one (Fig 4A, loop arrows). Based on the pathway dominance in each zone (Fig 4B), we confirmed our previous observations that commitment to a choice occurs as soon as the direct pathway becomes sufficiently dominant in an action channel (zones III and IV), while increased activation in the indirect and pallidostriatal pathways contributes to ongoing deliberation (zones II and V). This zone partition served as a succinct representation of the state CLAW by capturing the activity of the main pathway components at different phases of decision-making. As a summary of the above analysis, Fig 5 shows the temporal sequence of CBGT pathway activation patterns associated with different classes of decision trajectories along the zone CLAW.

This zone version of the CLAW provides some additional insights into the dynamics of decisions through CBGT networks. First, the dSPNs in both channels were fully activated in zone VI, suggesting a strong competition between the two options. The existence of this neutral zone indicates that the two outer zones were actually not absolute in their commitment: while most trials that reached these zones would remain committed to their initial choice, a few "indecisive trials" ultimately switched to the opponent action. This switch occurred due to the late ramping of the competing action channel, where delayed activation in its direct pathway became sufficiently influential to rapidly override the initial commitment and could lead to a decision reversal. Second, there were direct paths from zone II to the outer zones, representing occasional transitions from the inner CLAW to the outer CLAW that were only evident with grouping of states into zones (hence they did not appear in Fig 3A). These connections highlight the possibility of direct shifts from the early deliberation phase to action commitment, without a need to return to zone I for a fresh re-evaluation or to continue into a prolonged second deliberation phase. Along this shortcut, the dSPN firing rate rapidly increased above the threshold, suppressing the original ramping activity of the iSPN and GPeA, thereby preventing the trial from fully completing the deliberative process and correspondingly accelerating the decision-making. Third, the irreversible transition between the two deliberation zones (II and V) suggests that once the trial moved into a deeper, more fully activated phase of deliberation, the likelihood of resetting deliberation reduced, moving the process toward a concluding outcome and preventing indefinite uncertainty, albeit with a long decision time. Comparing zone V with zone VI, we see that both dSPN populations in zone VI were above threshold, while the activation pattern in zone V features an analogous balance across channels but for the indirect and pallidostriatal pathways (both iSPN, STN and GPeA populations were above threshold). Hence, the trials reaching in zone V could not turn back from this all-out battle, but rather waited to see which GPi firing rate would fall first. In this sense, although the dominant activity of the indirect and pallidostriatal pathways may temporarily delay the decision, the growing activation of the direct pathway in zone V ultimately ensured convergence to a choice. This interplay among pathways points out the network's ability to balance deliberation and commitment, maintaining functionality even in the face of competing influences. Overall, by categorizing CLAW states into key zones, we gained a clearer understanding of the critical transitions between deliberation and commitment, including certain shifts that were infrequent, but nonetheless enhanced flexibility in complex decision-making dynamics. The zone CLAW also helps us to appreciate the overall stability of decision-making processes despite the inherent variability present in neural firing activity, highlighting the robustness of the competition between CBGT subnetworks that effectively controls how incoming information leads to deliberation and decisions.

## CLAW and decision policies

Previous modeling studies, using two-choice decision tasks, have identified three low-dimensional control ensembles within the CBGT network. Each is associated with specific changes to the decision policy, defined in terms of parameter

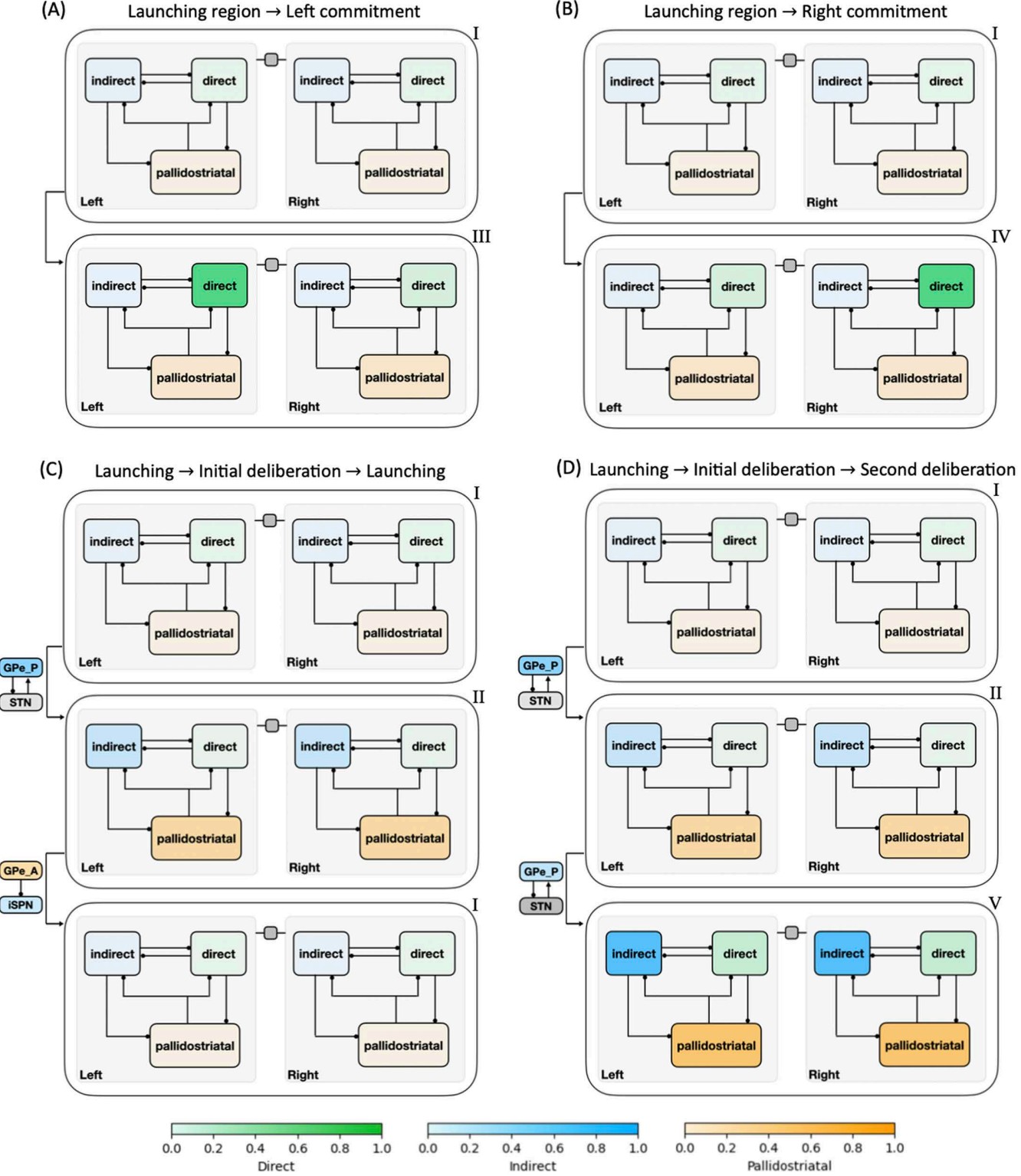

**Fig 5. Temporal activation of CBGT pathways for different decision trajectories along the CLAW.** The direct, indirect, and pallidostriatal pathways across both left and right channels are enclosed in boxes, with their activation strength colored in blue, green, and gold, respectively (color bars at bottom). Darker shades indicate stronger activation within each pathway. Nuclei that play a key role for some phase transitions are indicated near the corresponding arrows. **(A)** Launching region → Left commitment. **(B)** Launching region → Right commitment. **(C)** Launching region → Initial deliberation

→ Launching region. **(D)** Launching region → Initial deliberation → Second deliberation. The Roman numeral on the upper right corner of each pattern corresponds to the zone labels in Fig 4.

configurations of evidence accumulation models [22,23]. We replicated this analysis using our CBGT network architecture, with the aim of examining the relationship between different CLAW zones and decision policies. First, we gathered the behavioral features (decision times and choices) produced by the simulated CBGT networks and fit their distributions to the DDM using the Hierarchical Sequential Sampling Modeling (HSSM) toolbox (see https://lnccbrown.github.io/HSSM/ for more information on the HSSM toolbox). This step allowed us to derive sets of four key DDM parameters associated with the decision process for each network: boundary height $a$, drift rate $v$, onset time $t$, and starting bias $z$. We then applied canonical correlation analysis (CCA) to compute a low-dimensional mapping between correlated patterns in the space of CBGT network activity and the space of DDM decision parameters. For CBGT network activity, we considered two aspects of activity within each CBGT population: (1) global firing rates across the left and right action channels (i.e., sum of the firing rates in each region across both channels), and (2) bias in activity towards one action (i.e., difference in the firing rates of each cell population, between the left and right channels). From this analysis, we captured the first three pairs of component vectors that maximized the correlation between CBGT activity and DDM parameters. Fig 6 shows color-coded representations of the loadings from the vector $u$ consisting of the firing rate components and from the vector $x$ comprising the DDM parameter components from the CCA. See Control ensembles in the Methods section for more details.

Our analysis recovered three components that are nearly identical to the components found in prior work [22,23]. Comparing across columns, the first component is strongly associated with the drift rate $v$ and the between-channel differences in firing rates, consistent with the previously identified *choice* ensemble. Greater activity of cortical, striatal, STN, GPeA, and thalamic neurons in the left channel relative to the right channel, coupled with weaker activity in GPi and GPeP, corresponds to more positive drift rates toward the left decision. The second component, in contrast, loads heavily on the overall activity of corticothalamic systems as well as the direct pathway, with negative loading on both

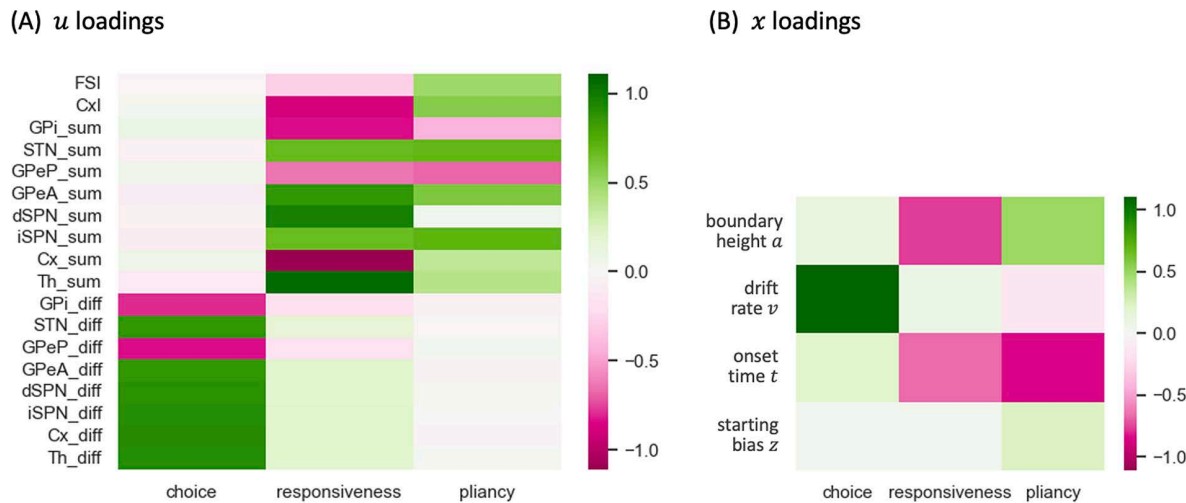

**(A)** $u$ loadings  **(B)** $x$ loadings

**Fig 6. Canonical loading matrices obtained by applying the CCA. (A)** Loadings of $u$, corresponding to CBGT firing rates. The subscript 'sum' refers to the sum of rates in a region across both action channels, while the subscript 'diff' refers to the rate in the left channel minus that in the right channel. **(B)** Loading of $x$, corresponding to DDM parameters. These three components are referred to as choice, responsiveness, and pliancy control ensembles, respectively.

boundary height *a* and onset time *t*. This pattern agrees with what had previously been called the *responsiveness* ensemble, because it modulates, in the same direction, both how quickly evidence evaluation begins and the level of evidence needed for a commitment to a decision and hence the overall response speed. The final component loads heavily on both the indirect and pallidostriatal pathways and shows a positive loading on *a* but a negative loading on *t*. This component matches well with the *pliancy* ensemble, reflecting the ease with which the evidence collected can be translated into a commitment to a decision. In our case, the choice ensemble corresponded to the strongest component, describing the most covariance between CBGT activity and DDM components, whereas in prior work it was the weakest of the three components [22,23]. Other than this ordering of components, our CCA results are highly consistent with prior work using different variants of the CBGT spiking network model.

With these control ensembles in hand, we next investigated how the dynamics of CBGT network activity during decisions tunes the resulting decision policy parameters. To this end, we translated the time series of CBGT firing rates to the time course of individual control ensemble engagement. Specifically, for each simulated trial we computed the change in the 18 CBGT firing rate measurements (see column labels in Fig 6A) between consecutive time bins, denoted by $\Delta F_k$ from the $(k-1)$-th to the $k$-th bin. Next, we projected $\Delta F_k \in \mathbb{R}^{18}$ onto the control ensemble space by mapping $W_k = \Delta F_k^T U$, where the components of the *u* loadings comprise the columns of $U$, such that $W_k \in \mathbb{R}^3$. Each component of $W_k$ represents how the instantaneous firing rate change from the $(k-1)$-th to the $k$-th time bin corresponds to a change in the activation, or *drive*, of one of the three control ensembles. Because each control ensemble impacts specific aspects of decision policies (Fig 6B), the time series of ensemble activations imply how the CBGT network activity dynamically modulates decision policies throughout the unfolding of individual trials. Finally, we grouped all trials into four types, categorized by their decision times and selected choices: fast left, fast right, slow left, and slow right. Given that decision times vary across trials, we aligned the trials in each group to the response time, as shown in Fig 7. In this way, we converted the temporal flow of neural activity into evolving control ensemble activations, that serve as direct reflections of dynamic decision policies encoded within the course of individual decisions. For a full description of this approach, see Control ensembles in the Methods section.

We observed that the main difference between fast choices (Fig 7A and 7C) and slow choices (Fig 7B and 7D) is the timing at which the drives began to change. For the fast trials, the choice drive showed a modest deviation from baseline, becoming positive for the left action and negative for the right action. This happened near the end of the decision time, marking the point at which the system began to show a directional commitment. However, the drive of the responsiveness and pliancy components rose earlier and more dramatically, around the midpoint of the decision process, indicating that the system acted early to modulate the response speed and the degree of evidence necessary for selection. For slow trials, the choice ensemble still changed a bit later than the other two ensembles. Yet, considering the much longer decision times on slow trials relative to fast trials (note the differences in time axis scales in Fig 7A and 7C vs. Fig 7B and 7D), the slow cases exhibited more similar timing across the three ensembles, with all exhibiting a gradual acceleration as the long decision process neared completion. In addition, the increase in the magnitude of the responsiveness and pliancy ensembles was notably smaller in the slow cases than that in the fast cases, while the changes in the choice ensemble closely overlapped (see the right insets in Fig 7). Thus, the comparison across decision types shows that the critical components regulating overall decision speed are responsiveness and pliancy.

We next considered how these control ensemble changes map to specific activity in the CBGT cellular populations. Consider the distinctions in the temporal patterns of control ensemble activation between fast and slow decisions in terms of the activity of specific CBGT pathway components. First, as shown in Fig 6A, the increase in responsiveness was associated with the overall activity in corticothalamic and direct pathways, and specifically, greater activity in thalamic and dSPN neurons along with weaker activity in cortical and GPi neurons. Second, the increase in pliancy corresponded to the overall activity in components of both the indirect and pallidostriatal pathways, reflected by the amplified activity in iSPN, STN, and GPeA neurons and attenuated activity in GPeP neurons. Third, the change in the choice ensemble was

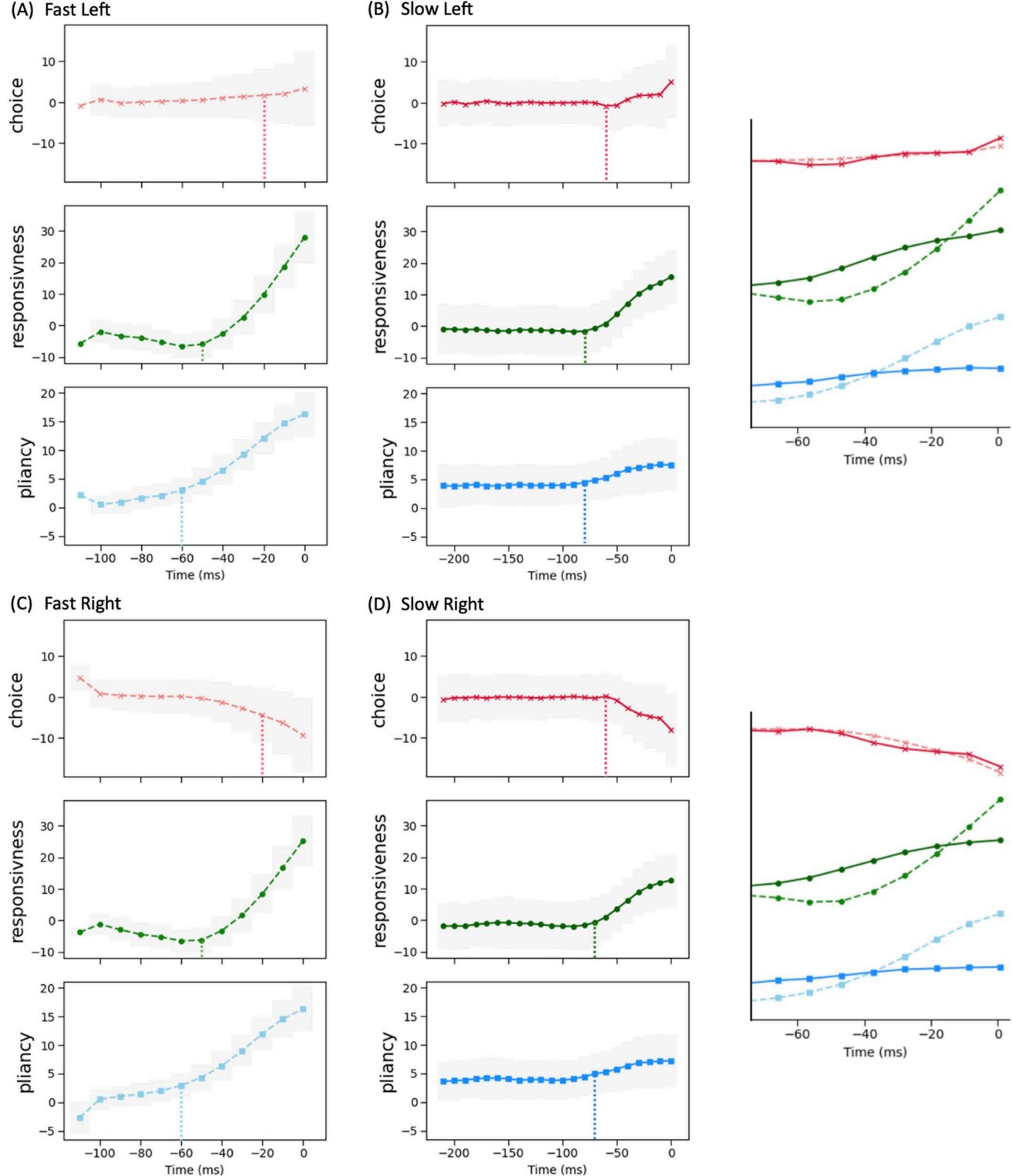

**Fig 7. Time evolution of control ensemble drives.** The percentage changes in each drive are averaged across all trials within each of four decision groups—**(A)** fast left, **(B)** slow left, **(C)** fast right, and **(D)** slow right, from the stimulus onset time to the decision time (set to be 0). Since each trial had a different decision time, we performed the averaging by aligning trials on their decision times, with the averages represented by dots connected by lines

(light dashed for fast and dark solid for slow). Shadowed areas represent the standard deviation for each group. The vertical dotted lines with colors matching the plotted curves mark the timing at which each control ensemble's drive shows a notable response. For fast and slow trials to the same choice, each inset at the right aligns the drives, zoomed in on the time range with pronounced changes (~75 ms), on the same scale for comparison.

driven by differential activity between the two channels in all pathways. From the timing difference observed from Fig 7, we inferred that a fast decision featured an early surge of overall CBGT activity before the emergence of between-channel differences. This seems counterintuitive, as one might expect that faster decisions would involve earlier differentiation between action channels, reflecting a more immediate commitment to one choice over the other. However, the delay in the emergence of between-channel differences aligns with the notion that fast decisions involve a stereotyped process, where the overall network activity assumes a state that is conducive to rapid implementation of whichever option subsequently becomes favored. As a result, only after the general decision approach had been established by the necessary components of the CBGT network, did the divergence of activity between channels become pronounced. This allowed for a smoother transition and potentially minimized unnecessary conflict or indecision between competing action channels. In contrast, the slow decision process was more deliberative, with prolonged competition and evaluation between options, including additional processing associated with transient pause, action switches, or suppression. These processes required the coordinated efforts of many CBGT nuclei, as we discussed in the CBGT activity and CLAW subsection in the context of the initial and second deliberation phases. Consequently, the turning point of the overall network activity for slow decisions occurred much later than in the fast cases and was associated with the emergence of a bias or preference towards one decision option.

In big picture terms, the temporal patterns that we observed in the control ensembles are consistent with our previous CLAW analysis showing that, at least in CBGT-driven decisions, there exist two stages—deliberation and commitment—to every choice. Note that, in contrast to the two extended deliberation phases that most fast trials bypassed (cf. Fig 4A, zones II and V), the deliberative stages of the fast trials occurred during the relatively brief time that the fast trials spent in the launching region (Fig 4A, zone I). During this time the rapid responsiveness and pliancy ensemble responses occurred (Fig 7A and 7C) as the CBGT network quickly organized itself for action. This analysis highlights a relationship between the contribution of the three control ensembles and the distinct phases of a decision, achieved through the corresponding patterns of dynamic interactions among CBGT pathways.

By exploiting the relation between control ensembles and DDM parameters, we next investigated how the dynamics of CBGT network activity during decisions can be formulated in terms of serially tuning the evidence accumulation process itself during the decision. For this step, and for each pair of zones $(i, j)$ we considered the change in the CBGT firing rate measurements associated with the direct transition from zone $i$ to zone $j$ across the subset of $p \leq 300$ networks that undergo this transition. Denote the firing rate change from zone $i$ to zone $j$ by $\Delta F_{ij} \in \mathbb{R}^{p \times 18}$. We projected $\Delta F_{ij}$ onto the control ensemble space using the $u$ loadings derived from the CCA and obtained $W_{ij} \in \mathbb{R}^{p \times 3}$. For each column of $W_{ij}$, a positive (or negative) element indicates that the sample aligns with the corresponding control ensemble in a positive (or negative) direction, and thus the parameters of DDM that have positive loadings in that control ensemble will increase (or decrease) as the CBGT activity progresses from zone $i$ to zone $j$. As both responsiveness and pliancy ensembles influence the boundary height and onset time, we finally projected $W_{ij}$ to the DDM space. Using the components of the $x$ loadings to form the columns of $X$, we computed $P_{ij} = W_{ij} X^T \in \mathbb{R}^{p \times 4}$ which relates the firing rate changes to changes in the DDM parameters. Hence, the median of all $p$ projections in each column of $P_{ij}$ is proportional to the overall change in the direction and magnitude of the corresponding decision policy parameter in the transition from zone $i$ to zone $j$. We also evaluated the extent to which the static CCA loadings remain informative for time-varying decision policy dynamics using a generalization-and-alignment test; the full analysis is reported in the Supporting information S2 Appendix.

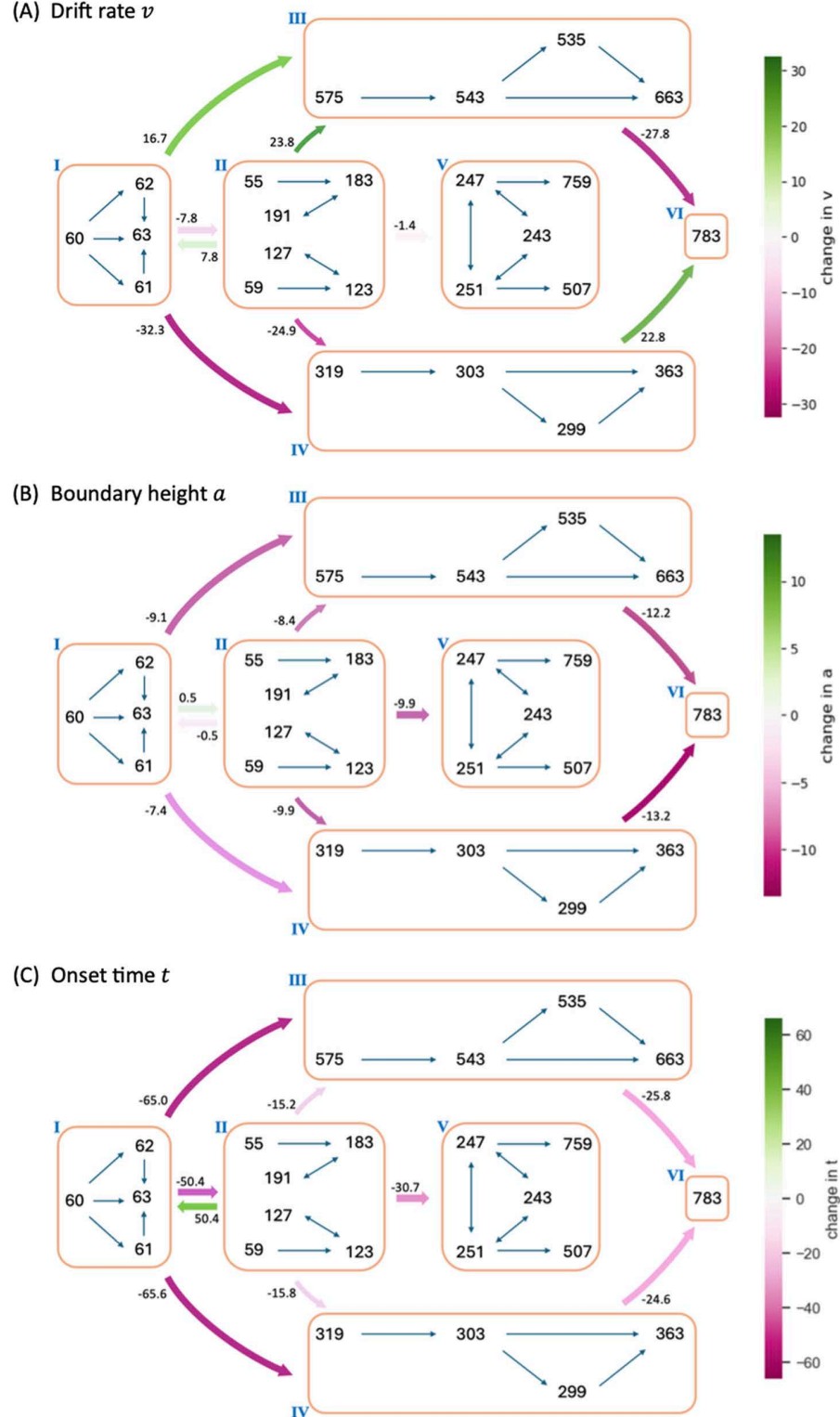

**Fig 8. Modulation of decision policy parameters by CBGT activity along the CLAW.** The arrows indicate variations of the individual DDM parameters (drift rate $v$, panel **A**; boundary height $a$, panel **B**; onset time $t$, panel **C**) associated with transitions between zones of the CLAW, computed based on the activity changes from each zone to the one immediately downstream from it. The magnitude of each variation is indicated by the coloring (see color bar) and the number at the start of each arrow, representing the percentage change in parameter values.

Fig 8 shows the influence of the CBGT network activity flow on each parameter of the DDM. The coloring and number of each arrow indicates the percentage change in the parameter values as the decision trajectory traveled between CLAW zones. Note that since all control ensembles had weak loadings on the bias factor $z$, we did not consider its associated changes. First, we observed that the modulation of the drift rate $v$ along the outer CLAW was strongly positive when committing to the left action (zones I/II to III) and strongly negative when committing to the right action (zones I/II to IV). These shifts in the drift rate resulted in a clear directional bias and a rapid approach toward the selection of the appropriate action. In contrast, $v$ was less affected when information flowed along the inner CLAW, leading to a much slower accumulation process regardless of the action choice. Reversion from the outer CLAW to zone VI, with equal likelihoods of the two decisions, was associated with changes in $v$ that countered the initial, commitment-related changes. Here we note that the pre-decision states appear to have been somewhat biased towards the left choice, as the magnitude of the modulation of $v$ for I→IV was twice as large as that for I→III. This indicates that the trajectories may not have needed as much of an increase in drift rate before reaching the left decision threshold. Correspondingly, the drift rate modulation for the initial deliberation transition I→II was small and negative, such that this progression counteracted the leftward bias and led to a more neutral exploratory state. Our results on the $v$ dynamics align well with the change in the activation pattern of CBGT regions across zones (Fig 4B), as the drift rate parameter was strongly associated with differential activity between left and right channels (see choice ensemble loadings in Fig 6)—these between-channel differences became significant as the network activity traveled along the outer CLAW, leading to dramatic changes in the drift rate, while they vanished almost completely along the inner CLAW and resulted in minimal drift rate changes.

Second, we observed a "boundary collapse" phenomenon, as each trial proceeded towards a commitment to a choice, as indicated by the pink arrows throughout Fig 8B. This collapse reflected a reduction in the amount of evidence needed for a choice to be made, which facilitated action commitment. An exception occurred at the transitions between the launching and initial deliberation phases (zones I and II), during which the boundary height remained almost unchanged, staying as high as when the evidence accumulation initially started. This invariance suggests a more cautious approach at the onset of decision-making, with the engagement of indirect and pallidostriatal pathway neurons blocking any imminent action. In contrast, progression to zone V lowered the decision boundary substantially, consistent with our previous analysis that the second deliberation ultimately ensured convergence to a choice. A closer inspection of the binarized firing rates in the two deliberation zones (Fig 4B) reveals that the neural populations in both channels were much more activated in zone V compared to zone II. This implies a dramatic alteration in both the responsiveness and pliancy ensembles and therefore the boundary height when transitioning to zone V.

Lastly, the onset time $t$ consistently shortened no matter which decision path was taken. Importantly, the onset time does not vary dynamically within a trial. Instead, the effect of varying $t$ through a CLAW path represents an updated estimate of when the evidence accumulation is presumed to begin, based on how far along the network state has progressed. For example, the drop in $t$ associated with the progression along the inner CLAW reflected the fact that the states in zone II and zone V corresponded to being farther along in the decision process than zones I and II, respectively. When the decision trajectory reached a later stage, e.g., zone V, the process "started" faster because it had already moved past the earlier stages, i.e., zones I and II, and became more "prepared". This does not mean that the actual onset time of the trial decreases. Rather, the model adjusts the prediction for the timing of accumulation onset as the decision process unfolds (see below and Fig 9 for more details).

Note that when we fit the DDM parameters to each CBGT network's outputs, we derived a single DDM parameter set that produced responses that most closely matched those of the CBGT network for the whole decision process. These parameters represent a static solution that captures the overall dynamics of decision-making across all zones and transitions for all trials progressing through the network. Then, to link the zone-specific activity patterns to the DDM parameters, we applied the same set of canonical vectors obtained by the CCA, as performed on the static network, to the firing rate changes between zones. Fig 9 provides a schematic illustration of this approach for the transitions from zone I to zones

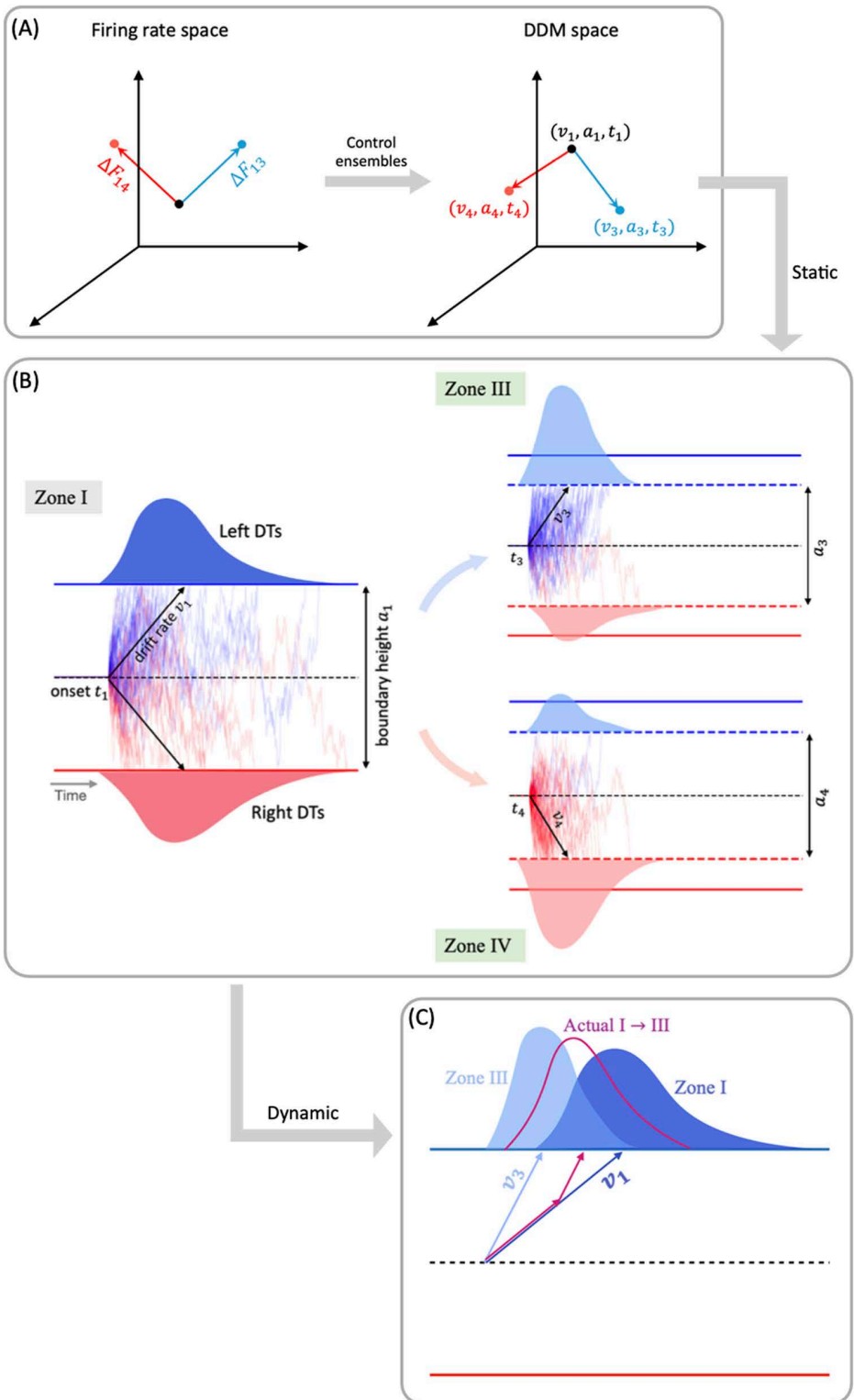

**Fig 9. Mapping changes in CBGT activity across CLAW zones to variations in DDM parameters. (A)** Firing rate changes during zone transitions are mapped to the DDM space through control ensembles. The vectors $\Delta F_{13}$ and $\Delta F_{14}$ represent the difference in the firing rate measurements from zone I to zone III (blue) and zone IV (red), respectively. These changes update the *static* DDM parameter set from $(v_1, a_1, t_1)$ to $(v_3, a_3, t_3)$ or $(v_4, a_4, t_4)$

. **(B)** The corresponding DDM behavior that fits the decision outcomes, assuming that CBGT activity stays in one zone (here, either zone I, zone III, or zone IV) over the entire decision process. **(C)** Prediction about the actual *dynamic* DDM behavior (magenta) as trials travel from zone I to zone III, the DT distribution of which is constrained by those corresponding to the static cases.

III and IV, respectively. The direct interpretation of this mapping is as follows: if the CBGT firing rates remained static throughout the decision process—resulting in fixed DDM parameters—and we then shifted to a different static set of firing rates (as if a new group of trials occupied a different zone for the entire decision), the canonical vectors would predict how the DDM parameters would need to change to stay consistent with the new activity pattern (Fig 9A and 9B). In reality, however, CBGT decision output arises from moment-to-moment changes in firing rates throughout each trial, rather than from a fixed set of DDM parameters applied to the entire trial. In other words, trials do not stay within a single zone for the whole decision process but instead transition dynamically between zones as the decision unfolds. Therefore, an accurate interpretation of our "dynamic DDM" analysis is to consider it as the movement of probability mass in the space of possible DDM parameters and DT distributions.

For instance, suppose that the averaged CBGT firing rates are those for zone I. If those stayed fixed, then we would obtain a specific set of DTs and choice probabilities consistent with a specific DDM (i.e., a specific set of fit evidence accumulation parameters). Similarly, consider another DDM parameter set that corresponds to the averaged CBGT firing rates for zone III. The difference in the static firing rates between zone I and zone III means that, relative to the first set, the DDM parameters and the probability mass of the DT distribution for the second set differ in a specific way (as depicted in Fig 9B). In the actual case of dynamic decision making, however, firing rate changes occur dynamically as the decision process evolves. Thus, for a decision involving a transition from zone I to zone III, the relevant dynamic decision policy must be intermediate between the two static cases, as illustrated by the magenta color in Fig 9C. For visualization, the figure only shows a possible variation in the drift rate, but the same idea applies to the boundary height and onset time. Although our current analysis does not deliver a specific trajectory of DDM parameters over time (i.e., $v(\tau), a(\tau), t(\tau)$, if time is parameterized by $\tau$), it allows us to estimate the impact of zone transitions on DDM parameters and hence to make a reasonable prediction about how passage through different zones will constrain the DT and choice distribution. Specifically, commitment-related zones induced significant changes in drift rate and boundary height, thus driving fast decisions, while deliberative zones resulted in minimal changes in these parameters and gave rise to slow decisions. Overall, our results demonstrate the dynamic nature of decision policies and their evolution across different decision-making scenarios.

## Discussion

One of the multifaceted functions of the CBGT circuit is regulation of the evidence accumulation process [3,16,58]. Yet, given that the synaptic architecture of the CBGT circuit is intricate, featuring complex interconnected feedforward, reciprocal, and feedback pathways [6], we know very little about how it regulates the flow of information on a moment-by-moment basis during an individual decision. Here we used an integration of multiple computational approaches, including generative spiking neural network models of the CBGT circuit and multiple data analytic techniques, to characterize the evidence accumulation process during individual decisions. Most critical to this was our CLAW framework, using which we were able to capture the flow of activity through CBGT pathways (Fig 3). We then grouped these states into zones to categorize key classes of decision trajectories (Figs 4 and 5). Further, by converting the CBGT dynamics into the time evolution of control ensemble drives, we identified that responsiveness and pliancy are the critical ensembles that determine the deliberation period of a decision (Fig 7). Finally, we recast our control ensembles results in terms of DDM parameters and investigated how CBGT activity tunes different aspects of decision policies as the decision process proceeds through different zones of the CLAW (Figs 8 and 9). Using this framework, we found that the noisy evidence accumulation process is

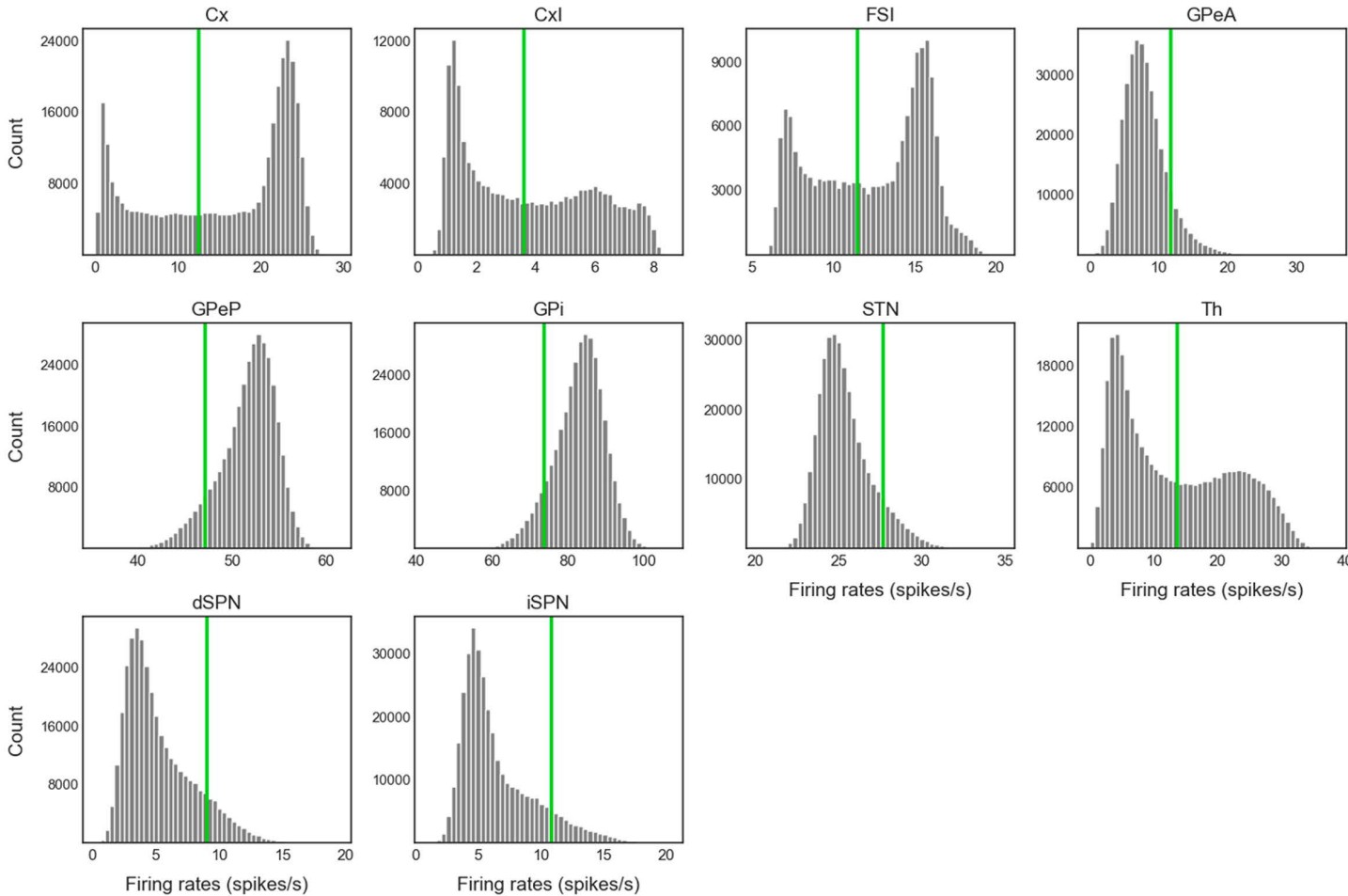

**Fig 10. Firing rate histograms for all trials up to decision times.** In each panel the vertical green line represents the binarization threshold for the firing rates. In unimodal histograms, the binarization threshold for GPeP and GPi was defined as the firing rate at 10% of the counts, with values above this threshold binarized to 1 and values below it binarized to 0, while for GPeA, dSPN, and iSPN the threshold was set at 90% of the counts. In bimodal histograms, the midpoint between the two peaks was defined as the binarization threshold.

functionally dissectible into two distinct phases, deliberation and commitment, each governed by competition and cooperation among distinct CBGT subnetworks. In particular, we found that the balance of overall activity across CBGT regions regulates the degree of evidence needed to make a decision early during the process, reflecting control of the deliberation stage. Eventually, a symmetry-breaking allows action-specific direct pathway components to kick in to trigger a transition from deliberation to commitment to a single choice relatively shortly before selection occurs. These findings suggest that the evidence accumulation process during a single decision is highly dynamic and is regulated by a complex interplay of multiple CBGT pathways that play distinct roles in controlling the flow of information during decisions.

Our results broadly confirm the growing perspective [3,4,17,44] that the direct and indirect CBGT pathways engage in a dynamic competition during decision-making, rather than merely functioning in isolation. Instead of a simple switch between "go" and "no-go" signals, the direct–indirect competition implements a decision by continuously balancing the drive for action with the need for deliberation. Our results add to this picture by including the pallidostriatal pathway and highlighting the ways that competition between pathways at key phases of the decision process determines its progression. Specifically, when the system adopts a responsive, pliant state early in the decision (Fig 7A and 7C), the direct

pathway associated with one option can gain strength and accelerate the network to action selection (outer CLAW; Figs 3 and 4). In other instances, the influence of the indirect pathway becomes dominant, pushing the system toward motor-suppressing states associated with prolonged deliberation (Fig 7B and Fig 7D). During this process, competition between the indirect and pallidostriatal pathways, as well as across action channels, determines the extent of deliberation (inner CLAW; Figs 3 and 4). This interactive nature of the CBGT pathways is also critical for encoding uncertainty and for adapting behavior in changing conditions. Our findings indicate that understanding the role of the CBGT circuit in shaping decision-making requires studying the intact dynamical system as a whole.

Previous recent work [22,23] introduced an "upward mapping approach" used to fit CBGT-driven decision outcomes to a normative DDM model, which led to the identification of three CBGT control ensembles (choice, responsiveness, and pliancy). Our work here shows that the control ensemble structure is preserved with the inclusion of the arkypallidal GPe neurons and demonstrates that the transition from deliberation and commitment emerges from the evolution of control ensemble activity (cf. Fig 7). Based on this control ensemble analysis, we translated the changes of CBGT network activity associated with transitions through CLAW zones into the evolution of individual DDM parameters, providing a time-dependent algorithmic interpretation of different decision strategies (see Figs 8 and 9). In particular, we observed that fast trials show an early and rapid reduction in decision boundary height while slow trials maintain a high decision boundary for a longer time. Unlike traditional DDMs, which assume that information is accumulated until a fixed threshold is crossed, our results align with the recently proposed notion of a "boundary collapse" [37,59–61], where the decision threshold dynamically adjusts in response to changing task conditions or sensory evidence. The distinction between fast and slow decisions in our analysis, however, emphasizes two contrasting decision strategies. On the one hand, the system may rely on an early and rapid collapse of the decision boundary to push the trial toward a timely action. This strategy would presumably be critical in situations that require swift decision-making under time pressure, allowing the system to optimize speed at the expense of deliberation. On the other hand, the system may devote an extended time to gradually lowering the evidence accumulation threshold, allowing for more evaluation of evidence before making a final commitment, thereby prioritizing accuracy over speed. Hence, the time-varying framework that we have derived serves as a more complete representation of how decision-making dynamics can be flexibly tuned to prioritize either speed or accuracy across diverse scenarios. One caveat is that our analysis used a static CCA mapping to infer time-varying DDM parameter dynamics. This step has been shown mathematically justifiable in our simulations at coarse temporal resolution (e.g., ≥10 ms bins), but becomes less accurate at very fine, near-instantaneous timescales (e.g., ≤1 ms) (see Supporting information S2 Appendix and S2 Fig), so our conclusions are more speculative in those cases.

Moreover, our results reveal that decision policies are highly dynamic during a decision, extending beyond boundary collapse to include other critical parameters such as drift rate and onset time, especially given the mixed opinions on the use of dynamic drift diffusion models [62]. We found that, as evidence accumulates, decisions involving outer CLAW paths manifest as a strong directional preference (zones III, V, and VI; Figs 4 and 8), driving the accumulation process rapidly towards a concluding outcome. In contrast, decisions that travel through inner CLAW (zones II and V) sustain a slower and more neutral information flow before committing to a choice. This difference in the drift rate dynamics between fast and slow decisions highlights the flexibility of the decision-making system in adjusting its approach to support varying strategies for making choices. Meanwhile, we observed similar time-evolving patterns of the choice ensemble drive, which modulated the drift rate parameter, as both fast and slow decisions neared completion (insets in Fig 8, red curves). This reflects the functional robustness of the system in terms of maintaining a stable stereotyped process for decision-making under different conditions.

The results obtained in this work also support the view that prototypic GPe neurons' connections to STN and GPi form an integral component of the classical indirect pathway, whereas arkypallidal GPe neurons [45] play a critical role in regulating upward information processing in the CBGT circuit [53,63–65]. Complementing experimental findings on the two subpopulations' distinct firing rate characteristics [48,66], our simulations and analysis suggest new insights into how

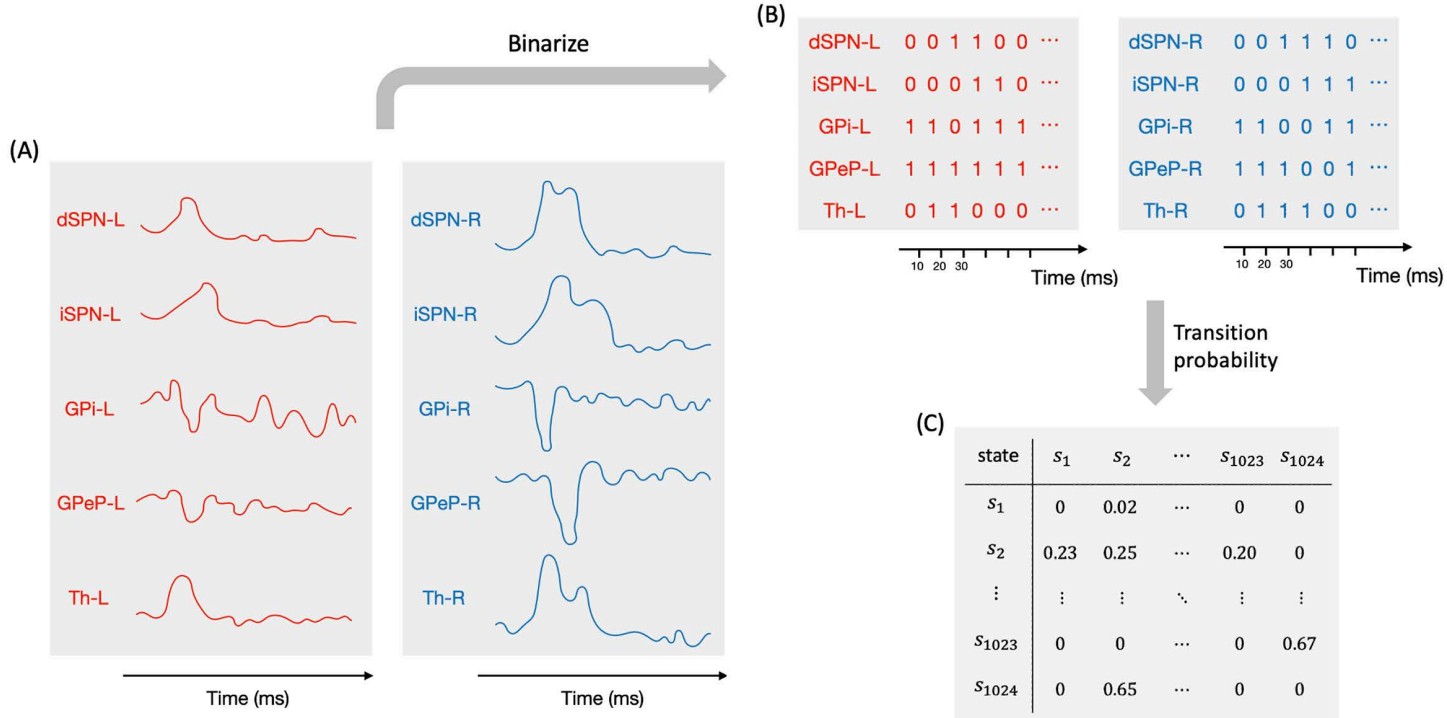

**Fig 11. Procedures for processing firing rate data. (A)** Populations of interest are specified, with the simulated temporal activity. **(B)** Firing rates within each population in each time bin (10 ms) are binarized into 1 (above threshold) or 0 (below threshold). **(C)** Transition probabilities are computed using the full set of binarized activity sequences.

the two GPe subpopulations may make divergent contributions to controlling the dynamics of individual decisions. As shown in Fig 2, the prototypic neurons fire at high rates during resting states, with many neurons decreasing their activity in response to synaptic drive. In contrast, the arkypallidal neurons maintain low baseline firing rates during rest, while they rapidly and robustly increase their activity upon stimulus onset. This divergence in their firing properties has distinct impacts on the associated basal ganglia targets for each cell type. For the prototypic neurons, a decrease in firing rate as a decision unfolds fits well with the classical understanding of the GPe's role within the indirect pathway, providing a disinhibition of STN and the basal ganglia output nuclei [13,19,41]. In our CLAW framework, the second deliberation phase (the loop among states 247, 243, and 251 in Fig 3A), characterized by zero binarized firing rates in one or both prototypic populations, essentially functions as a "brake" on action selection, allowing the system to terminate unwanted actions. This phase closely resembles the dynamics of the classical stop signal task, where the goal is to inhibit a planned action in response to a "stop" signal [67,68]. This similarity holds even though no trial in our simulations completely ceased or refrained from making a choice, with STN-GPeP subnetwork dynamics evolving such that the trial effectively paused, re-evaluated, and potentially switched between alternative options. Hence, this phase suggests that the prototypic GPe's brake-like function operates not only during explicit stop signals, but as a more general mechanism for temporal regulation during decision-making.

In contrast to prototypical GPe neurons, the arkypallidal GPe neurons do not project to the STN or other downstream targets of prototypic neurons. Rather they appear to exclusively innervate the striatum via ascending projections [69]. Due to their robust activation upon trial onset, they complement the function of the STN to cancel an action through their direct suppressive effect on firing at the level of striatum. Our analysis shows that in the early deliberation phase (the loop

among states 63, 55, 183, and 191 in Fig 3A), the STN acts in the first-tier stage to suppress the basal ganglia output nuclei and pause an incipient response (from 183 to 191). The arkypallidal neurons then become engaged during the second-tier stage to complete the blocking of a response by inhibiting the striatal regions (from 191 to 63). Note that the arkypallidal neurons receive comparatively weak inhibitory striatal inputs compared to their STN excitatory inputs [47,70], which allows for the recruitment of arkypallidal neurons into the process of resisting commitment when activated by STN. Our results support the emerging standpoint [48,63,69,71] that the GPe, with its heterogeneous collection of cell types, acts not merely as a passive way-station in the indirect pathway, but as an active control mechanism in regulating bidirectional information flow to facilitate flexible execution or blocking of responses [63].

Whether arkypallidal neurons predominantly target SPNs of the indirect or direct pathway remains unclear. It was observed in [72] that the inhibitory signals from arkypallidal neurons have a larger amplitude to iSPNs than dSPNs, with an approximate ratio of 2:1. This suggests a preferential influence over the indirect pathway. Other experimental studies suggested less pronounced bias and more balanced innervation of the two types of striatal populations [69,73]. Prior modeling work, on the other hand, suggested that a preferential influence of arkypallidal neurons onto dSPNs is crucial for regulating reactive stopping decisions [53]. In our model, in the early deliberation phase, the arkypallidal neurons effectively inhibit both pathways, suppressing the overall progression of information flow in the circuit. This manifests as non-preferential innervation of both types of SPNs by the arkypallidal neurons, subserving an activity pattern akin to an "all stop" signal to the striatum. In the transition from deliberation to commitment, we also found potential roles for inhibition from arkypallidal neurons to both iSPNs and dSPNs. Specifically, the inhibition from arkypallidal neurons to iSPNs in both channels disinhibits the associated dSPNs, facilitating their competition and ultimately leading to a specific response selection (e.g., $247 \rightarrow 759$ and $251 \rightarrow 507$ in Fig 3). Moreover, when a trial becomes stuck in a cycle of indecision – where arkypallidal neurons inhibit both iSPNs and dSPNs (as seen in the three-state reciprocal transitions during the second deliberation phase)—a reduction in dSPN inhibition in either channel can be crucial for breaking the deadlock and allowing the trial to progress toward a decision. Overall, this analysis reveals that the inhibition of striatal populations by arkypallidal neurons plays different roles at different stages of the decision-making process. Although the relative strengths of the inhibitory pathways from arkypallidal neurons to iSPNs and dSPNs may affect which aspects are dominant, our results suggest that both components contribute to the ongoing competition within the circuit and hence offer a potential reconciliation of prior experimental and theoretical work.

Methodologically, our CLAW analysis relies on binarizing the firing rate within each CBGT nucleus at each time step, using the binarized rates to define activity states, and tracing transitions between states over time. Such a set of states and transitions between them can be called a Boolean network [74]. The Boolean network framework has been heavily used in computational biology, especially to understand dynamics of complex cell signaling pathways [75–77]. The use of this framework has been far less prevalent in neuroscience [78]. Although a recent paper took this approach for the analysis of fMRI data [79], to our knowledge the use of Boolean frameworks for studying the dynamics and evolution of activity in a neural circuit is novel. Our work here highlights the utility of this approach. Harnessing this methodology in future studies could provide an alternative to mean field approaches and other forms of coarse-graining, with a distinct emphasis on discrete states and state transitions that are meaningful for cognitive functions.

Our work suggests several interesting directions for future explorations. We considered a pre-defined action selection task and did not include dopamine-driven corticostriatal plasticity. Given the critical role of dopamine in modulating both the direct and indirect pathway neurons of the striatum [80–84], incorporating plasticity would be expected to significantly alter the pattern observed in our CLAW diagram and will be a key future step. Moreover, we simply considered unbiased evidence for the two available choices, so a natural direction for the extension of this work is to introduce asymmetric evidence in plasticity studies. Another interesting avenue for future inquiry would be to consider the impact of perturbations, consisting of changes in reward contingencies (global) and optogenetic stimulation of specific cell populations (local), on network trajectories and decision outcomes. These extensions could deepen our understanding of how the CBGT circuitry

shapes decision processes in response to varying conditions, offering insights into potential therapeutic strategies for disorders related to basal ganglia dysfunction.

Since the CLAW states that we derived from the binarized firing rate data were defined in terms of the activity levels of CBGT populations, our simulation results delivered immediate biological interpretability. The insights from our results suggest several experimentally testable predictions about how, in unfamiliar decision-making tasks that have not yet been learned, distributed control of decisions flows through the CBGT network over time. At the most global level, we predict that especially fast decisions should feature a distinctive early surge in dSPN activity with an associated drop in GPi firing. On the other hand, enhanced iSPN activity and reduced GPeP activity without this dSPN surge should associate with shifts into slow deliberative behavioral states, and should be followed by increased STN and GPeA activity, with an increased level of variability in STN-GPeP circuit activity and intensified GPeA activity together manifesting as a prolonged deliberative phase. This STN-GPeP aspect of our results is consistent with previous modeling work suggesting roles for the STN in action delay [85] and for variable STN-GPeP dynamics in promoting exploration [19,86]. While the initial increase in iSPN activity and decrease in GPeP activity may predominantly arise in one action channel, the emergence of these changes widely across the relevant nuclei is predicted to signify an entry into a more prolonged, second deliberation phase with variability in the ensuing choices selected.

Optogenetics allows for targeted activation or inhibition of specific neuron populations or types. Our results predict that brief GPeP stimulation or iSPN inhibition early in the decision-making process will block entry into a deliberative state, allowing dSPN activity to take over and produce a decision. Consistent with the above discussion, we predict that dSPN stimulation and GPeP stimulation should both accelerate decisions; stimulation of these whole populations, at least in the absence of significant experience, should lead to balanced likelihoods of selecting available options. Our results also predict that stimulation of GPeA neurons should yield prolonged deliberation. In addition to informing future optogenetic experiments, the CLAW analysis can be applied to large-scale neural datasets to extract important insights into real brain decision circuits (see Supporting information S1 Appendix for the application to neural recording data of [87]). Finally, a novel prediction of our control ensemble analysis (Fig 7), in light of the composition of the responsiveness and pliancy ensembles (Fig 6), is that even with fast decisions, there should be a discernible rise in activity throughout much of the CBGT network, with the exception of GPeP and GPi, before the channel-specific dSPN surge that induces commitment to a choice. The distinction between fast and slow decisions is predicted to relate not to the presence or absence of this global rise, but rather to its time course and slope (starting earlier but with a more gradual rise for slow decisions) and to whether it is quickly followed by channel-specific dSPN engagement (fast decisions) or not (slow decisions). Overall, we have introduced a new framework, the CLAW, for analyzing the dynamic flow of activity through a neuronal network and have used it to extract novel insights and predictions about CBGT dynamics, including GPeA neurons, during individual decisions. We hope that these advances spur additional experiments and modeling related to the detailed time course of neuronal dynamics during the performance of cognitive functions.

## Methods

### CBGT network

In this work we simulated behavioral data using a CBGT network model. The network consists of 10 distinct populations of spiking model neurons: the cortical interneurons (CxI), the excitatory cortical neurons (Cx), the striatum which includes the D1- and D2-expressing spiny projection neurons (dSPN and iSPN, respectively) and the fast-spiking interneurons (FSI), the globus pallidus external segment, which contains prototypical and arkypallidal neuron subtypes (GPeP and GPeA, respectively), the subthalamic nucleus (STN), the globus pallidus internal segment (GPi), and the thalamus (Th). We considered two groups of neurons for each population, corresponding to the left and right action channels, except for the CxI and FSI, which are shared across both channels. At the start of each decision trial, a constant stimulus input to the cortical neurons in both channels was turned on. This input resulted in a nonlinear increase in cortical firing rates and therefore

the ramping dynamics across the striatal populations, which impacted downstream activity in the rest of the basal ganglia and thalamic neurons. A choice was declared to be made when the firing rate of the thalamic population within one action channel reached 30 Hz before the firing rate of the other thalamic population. Each neuron evolves according to a conductance-based integrate-and-fire-or-burst model. Individual neurons within the same population were connected to each other with a population-specific probability ($p$) and connection strength (or weight, $w$). The maximal synaptic conductance ($g$) for an existing connection (e.g., AMPA, NMDA, and GABA) was given by $g = w$, while it was given by $g = pw$ in a population-averaged sense. Connections between different populations were similarly characterized by probabilities and weights. These probabilistic connections introduced trial-to-trial variability even with the same CBGT network parameter settings. Fig 1A illustrates the connectivity within the network, and the complete model details can be found in [54].

We implemented genetic algorithms (see [23] for details) based on values of 13 selected synaptic connection weights to generate 300 different network configurations that each feature firing rates of all CBGT populations within experimentally observed ranges [22]. In addition, each network exhibited trial timeouts (when no action is selected within 1000 ms) on fewer than 1% of trials, and we checked to ensure that each network was cortico-basal-ganglia driven, with positive correlation between cortical and striatal activity. With each obtained network, we simulated 50 decision trials and gathered temporal firing rate data for all populations across the two action channels.

To unfold the noisy neural firing activity during the dynamic decision-making process, we time-binned the firing rates with $\Delta t$ = 10 ms, averaged over neurons in each population, and binarized them based on whether activity was above or below a certain threshold in each time bin. To determine the binarization threshold, we generated firing rate histograms for all trials up to decision times, which showed either unimodal or bimodal distributions (Fig 10). For unimodal histograms, the binarization threshold was set at the firing rate corresponding to either 10% (GPeP and GPi) or 90% (GPeA, dSPN, and iSPN) of the total counts, depending on whether the nonstimulated firing rate of each population was above or below its firing rate close to decision times, with values above this threshold binarized to 1 and values below it binarized to 0. For bimodal histograms, the binarization threshold was set at the midpoint between the two peaks. Next we considered $N$ = 10 populations of interest—dSPN, iSPN, GPi, GPeP, and Th, for both left and right action channels—and we defined the set $\{s_k\}_{k=1}^{2^N}$ of vectors in $\mathbb{R}^N$ as the base-2 representations of all integers from 1 to $2^N$. Each $s_k = [\sigma_{k1}, \sigma_{k2}, \cdots, \sigma_{kN}]$, where $\sigma_{kj} \in \{0, 1\}$ for all pairs ($k, j$), denotes the $k$-th unique state of the network. Consequently, we derived a sequence of states from the time-binned, binarized firing rates of every single trial. Using all of these state sequences, we computed the transition probabilities between all possible states. See Fig 11 for a sketch of the above procedures used to process the firing rate data. Finally, we built a chain for the decision dynamics (as depicted in Fig 3) based on the state transitions. Specifically, for each state $s_k$, we sorted the subsequent $n$ ($n < 2^N$) paths from largest to smallest. To focus on the most likely transitions, we retained only the top-ranked paths until a significant drop in the probability occurred—defined as a gap of more than 25% between the $p$-th ($p < n$) and the ($p + 1$)-th paths. We treated such a gap as a cutoff, and the remaining lower-ranked paths were considered to have low likelihood and were excluded from our analysis. This whole approach for the binarization and CLAW reduces firing rate variability and provides a more refined model for the decision dynamics.

### Control ensembles

We replicated the steps of identifying three control ensembles, as introduced previously [22]. First, we mapped the behavior of our CBGT network to the drift-diffusion model (DDM) by fitting the distributions of decision times and choices of the simulated trials, using the Hierarchical Sequential Sampling Modeling (HSSM) toolbox implemented in Python, and obtained the configuration of DDM parameters (boundary height $a$, drift rate $v$, onset time $t$, and starting bias $z$) corresponding to each of the 300 network configurations. Then, through the application of canonical correlation analysis (CCA) on trial-averaged firing rates $F_{all} \in \mathbb{R}^{300 \times 18}$ and DDM parameters $D_{all} \in \mathbb{R}^{300 \times 4}$, we captured pairs of linear combinations within each data set, given by loadings ($u, x$) $\in \mathbb{R}^{18} \times \mathbb{R}^4$, that yielded the maximal correlation across data sets. This analysis identified three such components associated with distinct decision factors, referred to

as choice, responsiveness, and pliancy. The canonical loadings that we obtained are shown in Fig 6; see [22,23] for additional details on the method.

We converted the time-binned series of CBGT firing rates to a discretized time series of individual control ensemble drives. To do so, for each trial we computed $\Delta F_k \in \mathbb{R}^{18}$, the difference of the averaged firing rates in the 18 model populations between time bins $k$ and $k-1$. Then, $\Delta F_k$ was projected onto the three control ensemble components obtained from the above CCA, via $W_k = \Delta F_k^T U$ where the $u$ loadings form the columns of $U$, such that $W_k$ contains three elements, each representing the change of the corresponding control ensemble associated with evolving from the $(k-1)$-th to the $k$-th bin. Note that $\Delta F_0$ was set to zero as a baseline, representing the initial state of firing rates before any changes in the decision process occurred. As a result, the firing rates at subsequent time bins ($k > 0$) were adjusted relative to this baseline. The time series of each element of $W_k$ across the trials in each of the four decision groups (fast left, fast right, slow left, and slow right) is displayed in Fig 7.

Following the same approach, we used the trial-averaged firing rate in each CLAW zone to predict the potential changes in DDM parameters as the CBGT activity evolves across zones. Here we considered the changes in firing rates between each pair of connected zones across all $p \leq 300$ networks that had trials visiting both zones, denoted by $\Delta F_{ij} \in \mathbb{R}^{p \times 18}$ from zone $i$ to zone $j$, and used these changes to compute $W_{ij} = \Delta F_{ij} U$ where $W_{ij}$ is a $p$ by 3 matrix. The median of each column of $W_{ij}$ indicates the modulation of the associated control ensemble driven by the transition in CBGT activity from zone $i$ to zone $j$. Finally, we projected $W_{ij}$ to the DDM parameter space via $P_{ij} = W_{ij} X^T$ where the $x$ loadings comprise the columns of $X$, yielding a $p$ by 4 matrix. The median of each column of $P_{ij}$ reflects the overall change in the corresponding DDM parameter from zone $i$ to zone $j$, as shown in Fig 8. Because our zone-resolved DDM analysis relied on the static CCA mapping, we evaluated the validity of this static-to-dynamic inference using generalization-and-alignment tests (see Supporting information S2 Appendix).

## Supporting information

**S1 Table. CLAW state details.** The table in Fig 3 includes the details of the left-related and neutral states in the upper half of the CLAW, while this table provides the information for all CLAW states, with each pair of left- and right-related states showing symmetry up to the swap of certain L and R channel binary values.
(PNG)

**S1 Appendix. Applying CLAW to neural recordings.** To compare the CBGT dynamics predicted by our model with real brain dynamics, we applied our CLAW framework to the neural recording data from [87]. Our observations (see S1 Fig) demonstrate that the dynamics predicted by our theoretical CLAW—such as the relationship between SNr/GPi suppression and decision speed, and the distinct influence of CBGT pathways on the decision course—are also reflected in large-scale neural recordings, supporting the potential of our framework for interpreting the dynamics of real brain decision circuits.
(PDF)

**S2 Appendix. Validation of static-to-dynamic neural–policy mapping.** We evaluated whether the static CCA loadings learned from trial-averaged activity (Fig 6) can support inference about dynamic decision policy changes (Figs 7–9), using a toy-model generalization-and-alignment test. In the toy model, static CCA axes generalized to coarse temporal changes when the underlying neural–policy mapping was stationary or slowly time-varying and shared across networks, but failed when the mapping was network-specific (panel A in S2 Fig). Consistent with the toy model results, our CBGT simulated data showed tight bootstrap intervals for canonical correlations and stable per-variable loading structures, and exhibited strong alignment for CLAW zone transitions (panels B–D in S2 Fig), supporting a shared neural–policy mapping across networks. Hence, our analysis of using the static CCA subspace to interpret bin/zone-scale dynamics is justified at the temporal scale relevant here (≥10 ms), even though it is not expected to capture moment-to-moment (≤1 ms) fluctuations.
(PDF)

**S1 Fig. Empirical CLAW of [87].** (A) The effect of SNr suppression on decision times (DTs). Bars represent the mean DT (± standard error) across trials, grouped by the percentage of zero binarized SNr firing rates during the decision time. Each color corresponds to a different mouse. (B) Firing rate histograms for SNr, GPeP, and GPeA neurons from mouse 3 across all trials up to the decision times. The vertical green line indicates the binarization threshold for each population. (C) Emprical CLAW for mouse 3, built upon the states of SNr, GPeP, and GPeA binarized activity. Numbers in boxes indicate the network states (e.g., "6: [1,1,0]" denotes state 6 where the binary firing rates of SNr, GPeP, and GPeA are 1, 1, and 0, repsectively). The transition probability from a current state to a subsequent state is indicated by the number near the arrow pointing from the current state. Numbers below states (except state 7) represent the probability of reaching decision from this state. The coloring of each state box corresponds to the mean DT of all trials that visit this state. (PNG)

**S2 Fig. Generalization of static neural–policy mappings across regimes.** (A) Toy model: alignment strength (maximum absolute correlation across canonical dimensions) between neural and DDM projections at three temporal resolutions (trial-averaged, zone/bin differences, and moment-to-moment differences) under stationary (blue), shared dynamic (green; 2/5/10 segments), and network-varying dynamic (red) mappings. Static CCA axes were fit on trial-averaged data and evaluated on held-out networks. Alignment is preserved for stationary and shared mappings at coarse scales, degrades at finer scales, and collapses for network-varying mappings. (B–C) CBGT bootstrap stability: network-level bootstrap refits of static CCA showing stable canonical correlations (panel B), and per-variable loading structures for neural and DDM variables (panel C) with 95% intervals. (D) CBGT zone differences: the toy-model train/test alignment analysis applied to CLAW zone-to-zone activity differences using static CCA axes learned from trial-averaged data.
(PNG)

# Acknowledgments

We thank Jyotika Bahuguna, Cati Vich Llompart, Eric Yttri, and all members of the exploratory intelligence group for their helpful comments on the this work.

# Author contributions

**Conceptualization:** Timothy Verstynen, Jonathan E. Rubin.

**Data curation:** Zhuojun Yu.

**Formal analysis:** Zhuojun Yu, Timothy Verstynen, Jonathan E. Rubin.

**Funding acquisition:** Timothy Verstynen, Jonathan E. Rubin.

**Investigation:** Zhuojun Yu, Timothy Verstynen, Jonathan E. Rubin.

**Methodology:** Zhuojun Yu, Timothy Verstynen, Jonathan E. Rubin.

**Software:** Zhuojun Yu.

**Supervision:** Timothy Verstynen, Jonathan E. Rubin.

**Validation:** Timothy Verstynen, Jonathan E. Rubin.

**Visualization:** Zhuojun Yu, Timothy Verstynen, Jonathan E. Rubin.

**Writing – original draft:** Zhuojun Yu.

**Writing – review & editing:** Timothy Verstynen, Jonathan E. Rubin.

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
