## [Decision Letter · Decision Letter 0]

5 May 2025

PCOMPBIOL-D-25-00501

How the dynamic interplay of cortico-basal ganglia-thalamic pathways shapes the time course of deliberation and commitment

PLOS Computational Biology

Dear Dr. Yu,

Thank you for submitting your manuscript to PLOS Computational Biology. After careful consideration, we feel that it has merit but does not fully meet PLOS Computational Biology's publication criteria as it currently stands. Therefore, we invite you to submit a revised version of the manuscript that addresses the points raised during the review process.

---

As you will see, both reviewers argue that further theoretical and/or experimental validation is needed to support your proposed neural interpretation of dynamic decision policies. Reviewer #2 specifically questions the computational interpretation of within-trial variations in decision-making parameters, such as drift rate and boundary height, which are typically used as static variables in the Drift Diffusion Model (DDM). Reviewer #1 is somewhat more demanding, in that he invites you to re-analyze some existing neural datasets to test some of the model's predictions. I expect you to provide a clear and argumented point-by-point response, and to modify the paper when necessary. While evaluating the pros and cons of complying with reviewers' comments, please remember that I will ask both of them to reassess your revised manuscript.

---

Please submit your revised manuscript within 60 days Jul 05 2025 11:59PM. If you will need more time than this to complete your revisions, please reply to this message or contact the journal office at ploscompbiol@plos.org. Please include the following items when submitting your revised manuscript:

We look forward to receiving your revised manuscript.

Kind regards,

Jean Daunizeau

Academic Editor

PLOS Computational Biology

Hugues Berry

Section Editor

PLOS Computational Biology

**Journal Requirements:**

**Reviewers' comments:**

Reviewer's Responses to Questions

Reviewer #1: This manuscript presents an interesting method of analysing the dynamics of neural activity during choice processes. This method is applied to an extended version of a previously published model of brain circuits involved in decision making, and reveals different possible trajectories of neural activity in the model during choice trials. The manuscript is clearly written.

Major:

The manuscript shows that the proposed method can be applied to analysis of simulated neural activity in a computational model, and provides insights into how this model behaves. However, for this work to have a broader impact, it would be useful to also show if the proposed method can be applied to analysing neural activity of biological neurons, and to investigate if the real brain decision circuits behave in a way predicted by the model. I feel these questions can be addressed thanks to availability of large datasets on neural activity obtained in choice tasks. For example, data from an influential study (Steinmetz, N.A., Zatka-Haas, P., Carandini, M. et al. Distributed coding of choice, action and engagement across the mouse brain. Nature 576, 266–273 (2019).) is freely available online at: https://figshare.com/articles/dataset/Distributed_coding_of_choice_action_and_engagement_across_the_mouse_brain/9974357?file=30626811

This study recorded activity “from approximately 30,000 neurons in 42 brain regions” including the regions simulated in the manuscript, during a choice task similar to that simulated in the manuscript. Therefore, I feel it would be insightful to test if the proposed method can be applied to such experimental data. It would be particularly useful to select recording sessions in which inserted probes crossed the regions included in the model. It would be interesting to also compare the dynamics revealed by analysis of the data with the dynamics of the model. I realize that some of the analyses of the model cannot be replicated in the data (e.g. it is not possible to distinguish if striatal recorded neurons are part of direct or indirect pathways), but even comparing dynamics across different parts of cortico-basal-ganglia-thalamic loop would be exciting.

Minor:

L.836: “path exhibited a gap of 25%” – I do not understand what it means. Could you please clarify?

Typo:

L.643: “Figs 4 8”

Reviewer #2: *Summary:*

In the present paper, the authors propose an innovative modeling framework to gain insights into the changes of configuration of the cortico-basal ganglia-thalamic (CBGT) network.

First, they generate a variety of CBGT network models producing plausible firing rate ranges, based on predefined pools of neurons informed by the existing literature. A novel feature of their approach lies in the identification of all possible network states—defined as unique configurations of active and inactive neuronal pools—and the analysis of transition probabilities between these states. From this, they extract distinct state trajectories, notably one associated with rapid responses and another corresponding to prolonged deliberation prior to decision commitment.

The authors then perform a cross-correlation analysis and replicate their previous findings, showing that variations in the global firing rates of the different neuronal pools of the models are related to behavioral decision policy along three primary dimensions: choice, responsiveness, and pliancy. These dimensions are derived from drift diffusion model (DDM) parameters fitted to the networks’ behavioral outputs. Finally, they relate time-resolved firing rate variations to these previously identified decision policy dimensions and to the underlying DDM parameters. Notably, firing rate changes at state transitions are associated with collapsing decision bounds, with higher drift rates observed along the “fast response” trajectory compared to the “long deliberation” trajectory.

Overall, I think this is an interesting paper, with an inventive approach for studying fine circuit-level dynamics and bridging them with the current understanding of decision processes at the computational level. However I have a significant concern (described below) regarding the computational interpretation of networks dynamics, which is in my opinion critical to address in order to include these results to the manuscript. Aside from this, I only have minor suggestions.

*Dynamic decision policy:*

On page 14, starting at line 377, the authors explain that the CCA loadings relating global firing rates to three combinations of DDM parameters (i.e., decision policy dimensions or “control ensembles”), are also used to estimate dynamic decision policy variations based on within-trial firing rate variations. The authors state: “Each component of Wk represents how the instantaneous firing rate change from the (k − 1)-th to the k-th time bin corresponds to a change in the activation, or drive, of one of the three control ensemble” (line 385-387).

Later (starting on page 16, line 451), the authors apply the same kind of analysis, this time projecting firing rate changes at state transition (rather than between time bins) into the decision policy dimensions, and eventually into the space of DDM parameters.

While the results presented in Figures 7 and 8 are compelling, I find the interpretation of within-trial variations in DDM parameters (or their linear combinations) to be problematic. These parameters describe distributions of behavior across trials, not time-varying variables evolving within a trial. Although some parameters, such as the boundary height or drift rate, could be interpreted as varying over time, doing so alters their original meaning, which was used to define the decision policy dimensions in the first place. For others, such as the onset time, I do not find any possible interpretation if they vary dynamically. I believe it is essential to explicit more the theoretical rationale for these analyses and to explain how within-trial changes in "control ensemble" activity should be interpreted in the context of DDM-based decision policy. Without such clarification, the computational meaning of these projections remains ambiguous.

That being said, even without this computational interpretation, the identification of separate state trajectories within the CBGT network is an interesting and valuable finding in itself, especially coupled with the thorough analysis of pathways activation.

*Minor / Readability suggestions:*

- Fig. 3A: State IDs written in black on a dark purple background are hard to read and to find when looking for them.

- Fig. 3C, 4B: These figures use the labels “D1” and “D2” to refer to D1- and D2-expressing spiny projection neurons, but the main text and the other figures all use the abbreviations “dSPN” and “iSPN”, which complicates the reading.

- Fig. 5: Adding the zone labels (e.g. “Launching region → Left commitment”) as a title for each subplot would improve readability.

- Fig. 6B: Using the DDM full parameter names (e.g. “drift rate” instead of “v”) or mentioning them in the legend would help understanding the figure independently of the main text.

- Fig. 7A, 7C: The colors are very pale, making the plots quite hard to read.

- Line 48: There is a typo (“reacitve” instead of “reactive”).

- As explicitly mentioned in the Methods section, the details of the procedure used to generate models of the CBGT network are described in two preprints and a paper freely available. However, as these models are central to the argument of the present paper, I think it could benefit from a few more details, clarifying for example the structure of inputs received by the network and the source of variability from trial to trial.

**Have the authors made all data and (if applicable) computational code underlying the findings in their manuscript fully available?**

Reviewer #1: Yes

Reviewer #2: Yes

PLOS authors have the option to publish the peer review history of their article (what does this mean? ). If published, this will include your full peer review and any attached files.

**Do you want your identity to be public for this peer review?** For information about this choice, including consent withdrawal, please see our Privacy Policy .

Reviewer #1: No

Reviewer #2: No

**Figure resubmission:**

**Reproducibility:**



---

## [Decision Letter · Decision Letter 1]

30 Sep 2025

PCOMPBIOL-D-25-00501R1

How the dynamic interplay of cortico-basal ganglia-thalamic pathways shapes the time course of deliberation and commitment

PLOS Computational Biology

Dear Dr. Yu,

Thank you for submitting your manuscript to PLOS Computational Biology. After careful consideration, we feel that it has merit but does not fully meet PLOS Computational Biology's publication criteria as it currently stands. Therefore, we invite you to submit a revised version of the manuscript that addresses the points raised during the review process. In particular, Reviewer 2 still has a significant point on the issue of static vs dynamic DDM parameters.

Please submit your revised manuscript within 30 days Nov 30 2025 11:59PM. If you will need more time than this to complete your revisions, please reply to this message or contact the journal office at ploscompbiol@plos.org. Please include the following items when submitting your revised manuscript:

We look forward to receiving your revised manuscript.

Kind regards,

Hugues Berry

Section Editor

PLOS Computational Biology

Hugues Berry

Section Editor

PLOS Computational Biology

**Journal Requirements:**

**Reviewers' comments:**

Reviewer's Responses to Questions

**Comments to the Authors:**

Reviewer #1: I would like to thank the Authors for addressing my suggestion so thoroughly.

Reviewer #2: I thank the authors for addressing my previous comments, and especially for clarifying the interpretation of some of their results. The motivation for focusing on dynamic DDM parameters is very clear, particularly in light of the new revisions. However, I remain concerned about the methodology used to link CBGT activity to these *dynamic* DDM parameters.

In my first review, I wrote: “Although some parameters […] could be interpreted as varying over time, doing so alters their original meaning, which was used to define the decision policy dimensions in the first place”. Let me clarify this point: my concern is that correlating CBGT network activity averaged over a trial with loadings of *static* DDM parameters does not ensure that similar relationships hold for near-instantaneous activity (Fig. 7, lines 388-408) or differences in activity between states (Fig. 8-9, line 469-487) when these are related to *dynamic* DDM parameters. Thus, the *static* loadings might not be usable to infer network drive for *dynamic* decision policies.

I think this assumption could be tested. For example, one could run a simple toy-model simulation in which a set of dynamic DDM parameters and their corresponding static parameters are generated. Cross-correlation analyses could then be performed separately:

- between trial-averaged activity and static parameters

- between within-trial differences in activity and dynamic parameters

If the resulting loadings are similar, it would support the use of *static* parameter loadings to infer drives for *dynamic* decision policies.

If such a simulation is impractical for reasons I have overlooked, I recommend explicitly noting in the Discussion that some caution is warranted when interpreting this part of the Results.

Apart from this point, I have no reservations regarding the rest of the manuscript, and I reiterate my view that it presents highly valuable findings and interesting methodology.

Minor points:

Line 590: “mulitple”

Line 654: “of of”

**Have the authors made all data and (if applicable) computational code underlying the findings in their manuscript fully available?**

Reviewer #1: None

Reviewer #2: Yes

PLOS authors have the option to publish the peer review history of their article (what does this mean? ). If published, this will include your full peer review and any attached files.

**Do you want your identity to be public for this peer review?** For information about this choice, including consent withdrawal, please see our Privacy Policy .

Reviewer #1: No

Reviewer #2: No

**Figure resubmission:**
---

## [Decision Letter · Decision Letter 2]

17 Feb 2026

Dear Dr. Yu,

We are pleased to inform you that your manuscript 'How the dynamic interplay of cortico-basal ganglia-thalamic pathways shapes the time course of deliberation and commitment' has been provisionally accepted for publication in PLOS Computational Biology.

Best regards,

Hugues Berry

Section Editor

PLOS Computational Biology

Hugues Berry

Section Editor

PLOS Computational Biology

Reviewer's Responses to Questions

**Comments to the Authors:**

Reviewer #2: I thank the authors for their thorough and thoughtful revisions, and I think they have done an excellent job addressing my concern. The additional analyses and supplementary material convincingly resolve the issue and, in my opinion, strengthen the manuscript.

I have no remaining reservations.

**Have the authors made all data and (if applicable) computational code underlying the findings in their manuscript fully available?**

Reviewer #2: Yes

PLOS authors have the option to publish the peer review history of their article (what does this mean? ). If published, this will include your full peer review and any attached files.

**Do you want your identity to be public for this peer review?** For information about this choice, including consent withdrawal, please see our Privacy Policy .

Reviewer #2: **Yes:** Juliette Bénon

---

## [Editor Report · Acceptance letter]

PCOMPBIOL-D-25-00501R2

How the dynamic interplay of cortico-basal ganglia-thalamic pathways shapes the time course of deliberation and commitment

Dear Dr Yu,

I am pleased to inform you that your manuscript has been formally accepted for publication in PLOS Computational Biology. Your manuscript is now with our production department and you will be notified of the publication date in due course.

With kind regards,

Judit Kozma
